

# 1 TROPOMI NO₂ for urban and polluted areas globally from 2019 to
# 2 2024

Daniel E. Huber[1], Gaige H. Kerr[1], M. Omar Nawaz[2], Sara Runkel[3], Susan C. Anenberg[1], Daniel L.
Goldberg[1]
[1]Department Environmental and Occupational Health, Milken Institute School of Public Health, George Washington
University, Washington, DC, USA
[2]School of Earth and Environmental Science, Cardiff University, Cardiff, United Kingdom
[3]National Center for Atmospheric Research, Boulder, CO, USA
*Correspondence to*: Daniel E. Huber (daniel.huber@gwu.edu)
**Abstract.** We present a global assessment of space-based urban nitrogen dioxide (NO₂) observation trends from 2019 to 2024
using annual and monthly mean tropospheric vertical column densities (VCDs) from the TROPOspheric Monitoring
Instrument (TROPOMI). Across 11,500 cities defined by the Global Human Settlement Layer-Settlement Model (GHS-
SMOD), we find population-weighted annual mean urban NO₂ VCDs declined between 2019 and 2024 in Asian (-17%),
European (-13%), and North American (−4%) cities, with seasonal decomposition indicating that  most of the annual changes
are driven by wintertime concentration decreases. South American (-2%) cities exhibited lesser population-weighted changes
on average, while African (+3%) cities experienced a gradual increase in NO₂. Over this timeframe, Tehran had the largest
NO₂ VCDs (>30 × 10$^{15}$ molecules cm$^{-2}$) and Seoul had the largest reduction (-40%). We further identify changes in NO₂ near
fossil fuel operations and note conflict-related changes in NO₂, highlighting the responsiveness of satellite NO₂ to certain
societal disruptions. We then calculate NO₂ VCD urban enhancements (VCD$_{ENH}$) by removing background concentrations
from urban signatures and compare VCD$_{ENH}$ to changes in nitrogen oxide (NOx) emissions from the Emissions Database for
Global Atmospheric Research (EDGARv8.1), to highlight regions with potential inventory discrepancies. We find VCD$_{ENH}$
and EDGARv8.1 NOx change at a similar rate from year to year in Europe and North America, with worse agreement in the
Global South. This work demonstrates the value in space-based remote sensing being an accountability agent for air pollution
emissions on a global scale and to identify changes in NO₂ in otherwise unmonitored regions.

## 25 1 Introduction

Nitrogen dioxide (NO₂) is a harmful air pollutant that originates from both anthropogenic and natural emissions sources,
including fossil fuel combustion, biomass burning, lightning, and soils (Dix et al., 2020; Jin et al., 2021; Schuman & Huntrieser,
2007; Huber et al., 2024), with fossil fuel combustion accounting for ~45% of total global nitrogen oxide emissions (Song et
al., 2021). Only a small amount of NO₂ is emitted from these sources directly, with nitric oxide (NO) being the primary
emissions product that quickly cycles to NO₂ in the presence of oxidants such as ozone (O₃) or peroxy radicals (HO₂ or RO₂).





The summed concentrations of NO and $NO_2$ are referred to as nitrogen oxides ($NOx = NO + NO_2$), as the concentrations of
NO and $NO_2$ are inherently linked. $NO_2$ is more commonly targeted by regulatory measures than NO, as it constitutes the
majority of atmospheric NOx concentrations and is linked to increased morbidity and mortality from long-term exposure,
particularly within urban environments (Chen et al., 2024). While NOx is commonly associated with health risks, the direct
association between NOx exposure and adverse health outcomes remains uncertain (Anenberg et al., 2022). Despite this, NOx
contributes to known harmful secondary pollutants, including $O_3$ and fine particulate matter.
$NO_2$ concentrations are measured using: (1) in-situ monitoring, e.g. chemiluminescence analyzers at the surface, or (2) remote
sensing instrumentation leveraging the unique spectral properties of $NO_2$, that absorbs light most efficiently in the visible
portions (405 – 465 nm) of the electromagnetic spectrum (Lamsal et al., 2015). The latter method relies upon spectrometers
detecting in the UV-Visible spectral range to infer $NO_2$ vertical column densities (VCDs), defined as the summed concentration
of $NO_2$ in a column from the surface to an upper limit of the atmosphere, with the tropopause often used as the upper limit.
Spectrometers have been used to measure $NO_2$ VCDs from ground-level directed upward, from aircraft directed downward, or
from space-based satellites directed downward, including from the TROPOspheric Monitoring Instrument (TROPOMI)
onboard the Sentinel-5P satellite (Herman et al., 2009; Fishman et al., 2012; Veefkind et al., 2012).
The earliest space-based spectrometers detecting $NO_2$ were flown on low-earth polar orbiting satellites, and were launched
within the mid-1990s to mid-2000s. These include the Global Ozone Monitoring Experiment (GOME; Burrows et al., 1999)
and GOME-2 satellites, the SCanning Imaging Absorption spectroMeter for Atmospheric CHartographY (SCIAMACHY;
Bovensmann et al., 1999) and the Ozone Monitoring Instrument (OMI; Levelt et al., 2006). The data collected using these
instruments provided unique insight into atmospheric chemistry and composition across the globe, including in mostly
unmonitored regions. OMI, launched in 2004, provided $NO_2$ VCDs at a spatial resolution of 13 x 24 $km^2$ at nadir and has
remained operable for more than two decades at the time this was written, providing a valuable long-term record of $NO_2$
globally. OMI remained the highest resolution space-based $NO_2$ product until TROPOMI launched in 2017, which ultimately
provided $NO_2$ VCDs at a spatial resolution of 3.5 x 5.5 $km^2$ at nadir. Observations at this resolution facilitated the evaluation
of satellite $NO_2$ at previously unprecedented spatial scales, including at the intra-urban level (Goldberg et al., 2021; Goldberg
et al., 2024).
$NO_2$ trends have been characterized in urban and broader environments using space-based instruments. Earlier satellite studies
used the GOME and SCHIAMACY satellites to identify increasing $NO_2$ VCD trends in China from the mid-1990s to the mid-
2000s (Richter et al., 2005; Stavrakou et al., 2008; Van der A et al., 2008), driven primarily by economic growth and
industrialization. Later studies, incorporating OMI observations, highlighted further increases in China through the early
2010s, with VCDs and satellite-inferred surface concentrations steadily declining since (Miyazaki et al., 2017; Jiang et al.,
2022). Europe has exhibited steady $NO_2$ VCD declines since the start of the satellite $NO_2$ record (Richter et al., 2005; Krotkov
et al., 2016; Duncan et al., 2016), driven largely by the implementation of various emissions control technologies. In the United
States, $NO_2$ concentrations increased through roughly 2005, then decreased substantially through the early to mid-2010s



(Lamsal et al., 2015), with VCD decreases more gradual since, in part due to an increased influence from regional background
$NO_2$ levels (Jiang et al., 2018; Dang et al., 2023). Additionally, numerous studies have highlighted the influence that the
COVID-19 pandemic had on $NO_2$ globally, with most regions exhibiting broad $NO_2$ decreases in 2020 during numerous
lockdowns and subsequent, regionally-distinct rebounds in emissions (Lonsdale et al., 2023; Fisher et al., 2024).
Satellite studies have been used to characterize trends within the urban environment specifically, using different methods to
characterize the urban extent. Geddes et al. (2016) used GOME, SCIAMACHY and GOME-2 oversampled to a 0.1° x 0.1°
grid to highlight $NO_2$ VCD trends globally, as well as in select urban areas, with the urban region defined as the surrounding
~ 200 km x 200 km. Fioletov et al. (2022) and Fioletov et al. (2025) used urban density from the Gridded Population of the
World (SEDAC, 2017) as a proxy for the extent of the urban environment to identify changes in urban NOx emissions.
Anenberg et al. (2022) used urban boundaries provided from the 2019 version of the Global Human Settlement Layer-
Settlement model (GHS-SMOD) to evaluate $NO_2$ trends from 2000 – 2019 using surface $NO_2$ estimates derived from
TROPOMI $NO_2$ and a land-use regression model.
Here, we use TROPOMI tropospheric $NO_2$ VCDs to quantify general $NO_2$ trends globally from 2019 to 2024, with a particular
focus on urban areas. The urban boundaries are defined by the 2023 version of GHS-SMOD, which provides urban cluster
boundaries for all urban regions globally. We evaluate urban $NO_2$ trends against a NOx emissions database, and evaluate the
influence of different seasons on annual trends. We additionally note changes in select oil, gas, and other mining regions,
which exhibit the largest changes globally outside of urban areas. This study represents the first detailed global-scale analysis
of urban TROPOMI $NO_2$ trends from 2019 to 2024. Our findings illustrate how $NO_2$ responded to specific societal events
during this timeframe, such as the impact of clean air policies, population migration away from urban areas due to war, the
increased demand for fossil fuels and rare-Earth minerals, and the emergence and waning of a global pandemic. Furthermore,
by directly linking observed $NO_2$ urban enhancements with NOx emission inventory data from the updated EDGARv8.1, our
work provides valuable insights into regions where emissions inventories align closely with observations, as well as areas
exhibiting potential inventory discrepancies. This work underscores the critical value of satellite-derived $NO_2$ as a tool for
urban air quality assessment and emissions management.
**2 Data and Methods**
**2.1 Global Human Settlement Layer Urban Cluster Boundaries**
The Global Human Settlement Layer-Settlement Model (GHS-SMOD; Schiavina et al., 2023) is a dataset developed by the
Joint Research Centre of the European Commission containing spatial boundaries and population estimates for all urban areas
globally with a population of at least 50,000, which can be used to subset gridded or spatially-disaggregated data for any built-
up area on Earth. GHS-SMOD uses satellite remote sensing to identify the spatial extent and boundaries of all cohesive built-
up areas globally at a spatial resolution of 1 $km^2$, with each separate, cohesive built-up area referred to as an "urban cluster".



In this study, we use the terms "urban cluster" and "city" interchangeably, although we note that GHS-SMOD urban clusters
do not necessarily align with administrative city boundaries. The 2023 version of GHS-SMOD provides boundaries for
approximately 11,500 urban clusters, along with population estimates for the year 2020 (Fig. S1). We note that GHS-SMOD
urban clusters do not reflect the traditional boundaries of individual cities as we may understand them, and as such, GHS-
SMOD urban clusters can span multiple cities, regions or even countries. For example, the urban cluster encompassing San
Diego, California includes the city of San Diego, but also the adjacent surrounding suburbs, as well as the entirety of Tijuana,
Mexico (Fig. S2). In such cases, attribution of an urban cluster to one particular city is not possible.
We use the GHS-SMOD boundaries to subset monthly- and annually-averaged satellite $NO_2$ column concentration data for all
urban clusters, as described in Section 2.2.1.
**2.2 TROPOMI $NO_2$ vertical column densities**
The TROPOspheric Monitoring Instrument (TROPOMI) is a pushbroom spectrometer on board the Sentinel-5P satellite
traveling in low earth orbit, with approximately one overpass each afternoon (Veefkind et al., 2012). Launched in October,
2017, TROPOMI detects radiation in spectral bands ranging from the ultraviolet to shortwave infrared to infer concentrations
of various atmospheric constituents, including nitrogen dioxide ($NO_2$), which is best inferred from the near-UV and visible
portions of the spectrum. We use Level 3 monthly- and annually-averaged TROPOMI tropospheric $NO_2$ vertical column
densities (VCDs) on a 0.1° global grid (Goldberg, 2024), which were created by oversampling daily Level 2 TROPOMI $NO_2$
VCDs, with a spatial resolution of 3.5 x 5.5 $km^2$ at nadir, derived from version 2.4+ of the European Space Agency retrieval
algorithm (van Geffen et al., 2022). The TROPOMI $NO_2$ data used in this study span six full calendar years from January 2019
to December 2024 (Fig. 1); we use the RPRO version from 1 January 2019 – 25 July 2022 and the OFFL version from 26 July
2022 – 31 December 2024. On 7 September 2024 there was an update of the surface reflectivity assumptions and on 16
November 2024 there was an update to the cloud retrieval, both of which induce a small positive step change in the data, but
likely does not meaningfully affect the 2024 annual average.



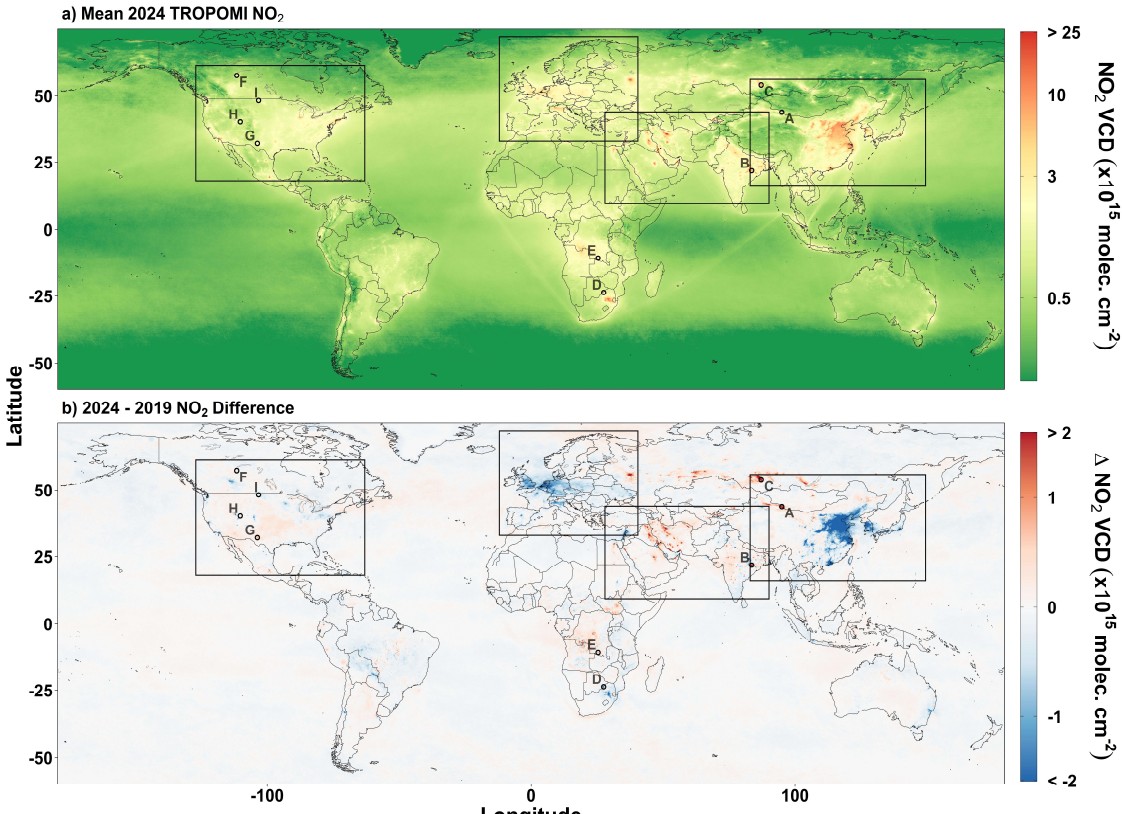

**Figure 1: (a) Global 2024 annual average NO₂ VCDs colored on a log-scale and (b) the difference in VCD from 2019 to 2024 colored on a symmetric log-scale. Points labeled A-I correspond with locations of oil, gas and mining operations highlighted in Fig. 3. Boxes indicate select focus regions in Section 3.**

## 2.2.1 Quantifying average TROPOMI NO₂ VCDs for GHS-SMOD urban clusters

For each urban cluster, we subset the oversampled TROPOMI data for grid cells that are located within 0.1° of the urban cluster boundary. For most cities, this results in approximately 20-25 grid cells, depending on the extent of the individual cluster. Given that the spatial resolution of GHS-SMOD is roughly an order of magnitude finer than that of the oversampled TROPOMI data (1 km vs. 0.1°) we interpolate the subsetted TROPOMI data to the 0.01° × 0.01° resolution of GHS-SMOD using a nearest neighbor approach We then calculate an area-weighted average of interpolated grid cells that have a grid cell center falling within the urban cluster boundary (Fig. S2). This approach allows for the portions of oversampled 0.1° × 0.1° grid cells that may not be centered within an urban cluster boundary, but that still overlap with a cluster, to be accounted for within the average NO₂ column estimate.





To evaluate the changes in VCDs for broader regions, e.g. countries containing multiple urban clusters, we can calculate a
population-weighted average VCD, taking into account varying population sizes in different urban clusters.
$VCD_{PW} = \frac{\sum_i(POP_i \times VCD_i)}{\sum_i(POP_i)},$  (1)
In Eq. 1, $VCD_{PW}$ represents the population-weighted VCD for a given country, $POP_i$ represents the 2020 GHS-SMOD-
estimated population for a given urban cluster $i$, and $VCD_i$ represents the mean $NO_2$ VCD for $i$.

**2.3 Accounting for background NO₂**

To account for changes in upwind background $NO_2$ concentrations that may influence urban $NO_2$ VCDs, we quantify an urban
$NO_2$ enhancement.
$VCD_{ENH} = VCD_{UC} - VCD_{BG},$  (2)
In Eq. 2, $VCD_{ENH}$ is the urban $NO_2$ VCD enhancement, $VCD_{UC}$ is the $NO_2$ VCD within each urban cluster as described in
Section 2.2.1, and $VCD_{BG}$ is the background concentration for an urban cluster. We define $VCD_{BG}$ as the 10[th] percentile of
$NO_2$ VCDs extending 0.5 degrees in any direction from an UC boundary (de Gouw et al., 2020).

**2.4 EDGAR NOx emissions**

We use version 8.1 of the Emissions Database for Global Atmospheric Research (EDGARv8.1; Crippa et al., 2024) to evaluate
NOx emissions. EDGAR provides summed total and sector-specific NOx emissions at 0.1° × 0.1° spatial resolution globally.
EDGAR NOx emissions include contributions from energy generation, industrial sources, transportation, residential sources
and agriculture. EDGAR emissions are produced using a bottom-up method that combines activity data together with sector-
specific emissions factors to produce gridded annual emissions. Similar to the handling of TROPOMI data (Sec. 2.21), we use
GHS-SMOD to quantify mean NOx emissions for each urban cluster.

**3 Global TROPOMI NO₂ vertical column densities from 2019 to 2024**

The following subsections describe the $NO_2$ VCDs and trends in four global subregions: Asia and Oceania, Africa, Europe,
North and South America.

**3.1 Asia and Oceania**

North and East China, one of the most populated regions globally with approximately 11% of the 1000 largest GHS-SMOD
cities, produced the broadest continuous expanse of 2024 annual mean $NO_2$ VCDs at or above 5 x $10^{15}$ molecules cm$^{-2}$ (Fig.
2a). Despite this, substantial decreases were observed in this region from 2019 to 2024 (Fig. 2b). While $NO_2$ concentrations
had already been decreasing in China prior to 2019 (Liu et al., 2016; de Foy et al., 2016), the decrease accelerated after the



onset of the COVID-19 pandemic, coinciding with reduced emissions during numerous lockdowns throughout the country
from 2020 to 2022 (Zheng et al., 2021; Ma et al., 2023; Zhao et al., 2024). The decrease in $NO_2$ also coincided with general
Chinese government policies directed at reducing emissions, including stricter emissions controls for industrial sources, energy
generation and the transportation sector (Shi et al., 2022; Li et al., 2024). Few regions in China experienced increased VCDs,
with the most notable increases occurring outside of major urban areas. The most substantial increase in VCD over China
through 2024 was observed in the sparsely-populated Santanghu Basin (Fig. 3a), a region in eastern Xinjiang Province with a
relatively nascent coal mining industry (Zhang et al., 2018; Liu et al., 2018). Annual mean $NO_2$ VCDs in the basin increased
by 1.9 x $10^{15}$ molecules $cm^{-2}$, or +172%, from 2019 to 2024. The expansion of mining operations is clearly evident in visible
satellite imagery (Fig. S3).

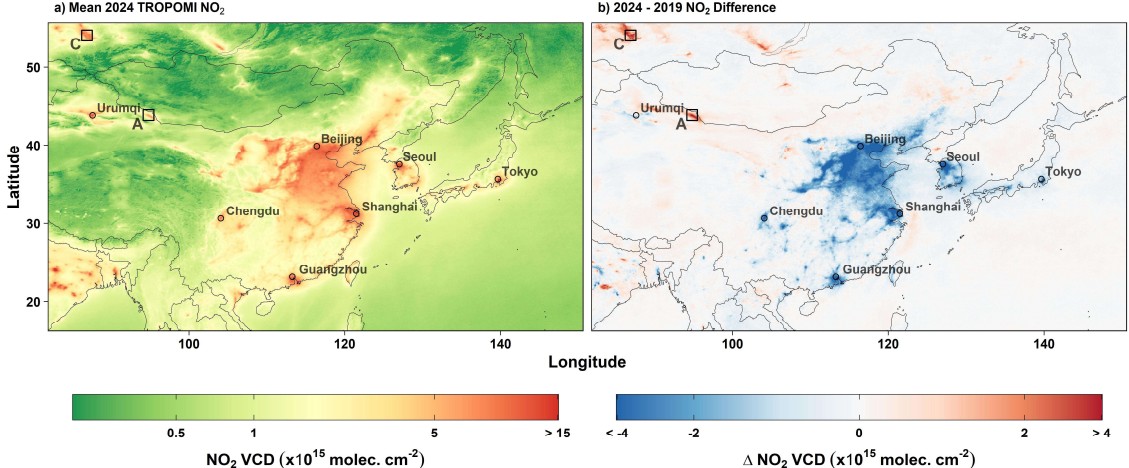


**Figure 2: (a) Mean 2024 TROPOMI $NO_2$ VCDs and (b) relative changes in TROPOMI VCDs for from 2019 to 2024, centered on**
**East Asia. Labeled black squares indicates the locations of mining regions highlighted in Fig. 3.**


In India, elevated $NO_2$ near numerous coal-fired power plants and coal mines is a common feature (Panda et al., 2023),
evidenced by the many apparent point sources in the 2024 annual average TROPOMI VCDs throughout the country (Fig. 4a).
$NO_2$ VCDs increased at many of these points sources from 2019 to 2024 (Fig. 4b), suggesting an increase in emissions from
energy production and use. The largest regional increase in VCD anywhere in India from 2019 to 2024 (+2.1 x $10^{15}$ molecules
$cm^{-2}$; +37%) was observed in the Ib Valley in northwestern Odisha state (Fig. 3b), a region with multiple surface coal mines
and coal-fired power plants (Varma et al., 2015). $NO_2$ VCDs near numerous other coal mines and power plants throughout
India exhibited changes, but $NO_2$ VCD increases were more prevalent than decreases. Of the major urban regions in India, the





largest decreases from 2019 to 2024 were observed in New Delhi (-1.6 x $10^{15}$ molecules cm$^{-2}$; -18%) and Mumbai (-1.0 x $10^{15}$
molecules cm$^{-2}$; -15%).

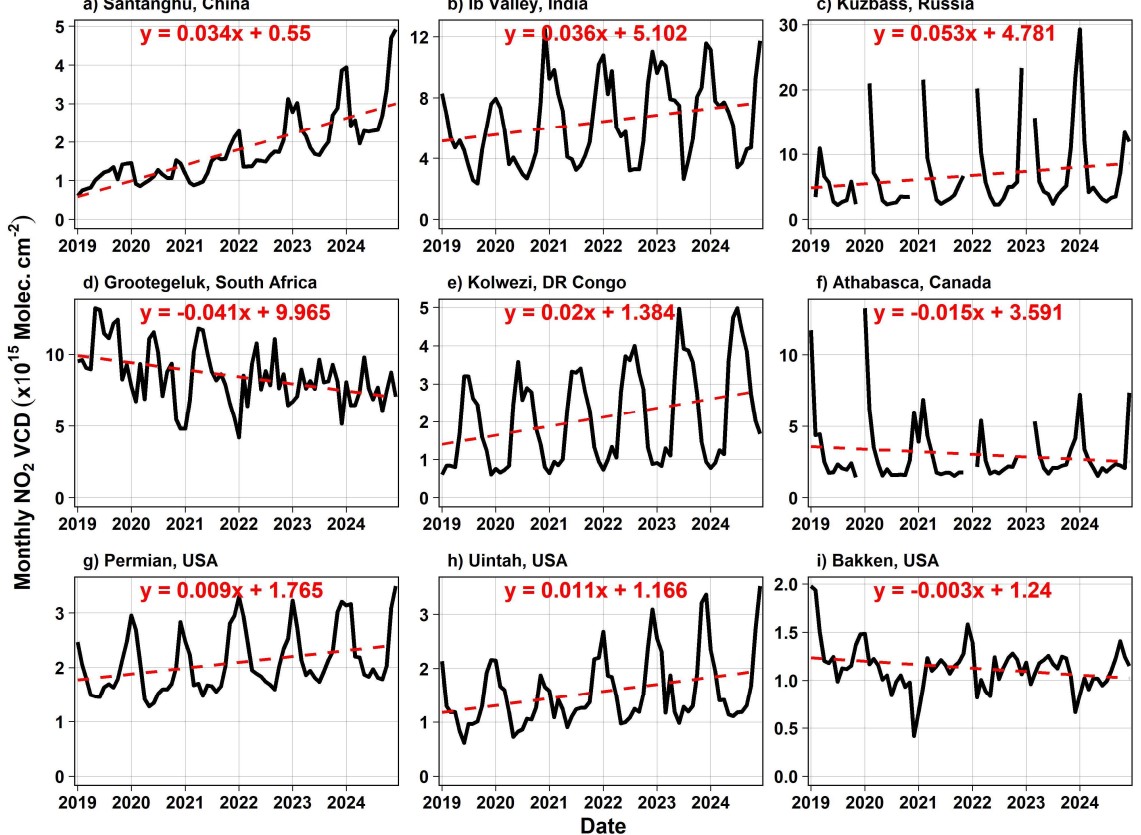

**Figure 3: Monthly time series of NO$_2$ VCDs over select oil, gas, and other mining regions. Black lines denote monthly mean VCDs, and red lines represent trends characterized by ordinary least-squares regression for each site. The slope of each trend line represents the change in NO$_2$ VCD per month, with the y-intercept representing the intercept for January, 2019. Months with missing data lacked quality-assured TROPOMI observations. Note the differing y-axis extents for each panel.**


Urban regions in Middle Eastern countries experienced some of the highest NO$_2$ VCDs globally in the TROPOMI record. Near
the Iranian capital of Tehran, 2024 annual average NO$_2$ VCDs of individual grid cells exceeded $40 \times 10^{15}$ molecules cm$^{-2}$ (Fig.
4a), the highest urban annual average among all global cities. Much of the Middle East exhibited substantial increases in
population-weighted, urban NO$_2$ VCDs from 2019 to 2024, most notably in regions of Saudi Arabia (+5%), Iraq (+18%), and
Iran (+10%), with broad increases that extend beyond the urban environment. One of the most salient VCD decreases in the



Middle East occurred in Lebanon (-39%), coinciding with the country's severe economic and financial crisis that began in late
2019 (Harake et al., 2019). VCD decreases through 2024 were particularly stark in the Lebanese capital Beirut (-6.7 x $10^{15}$
molecules cm$^{-2}$; -37%).  Additional Middle Eastern countries that exhibited decreased urban NO$_2$ VCDs through 2024 include
much of Israel (-27%), Kuwait (-5%), Qatar (-17%), and Afghanistan (-13%).
Nearly all urban regions in eastern Russia (Siberia) exhibited increased NO$_2$ VCDs, as did regions coinciding with known
mining operations. In the Kuzbass Region of Siberia, one of Russia's largest coal mining regions, annual mean VCDs increased
by 2.4 x $10^{15}$ molecules cm$^{-2}$ from 2019 to 2024, representing a 58% increase (Fig. 3c). A previous study identified a correlation
between space-based NO$_2$ observations and regional coal production in the Kuzbass region (Labzovskii et al., 2022), providing
relevant context for the observed VCD increases.
Other notable changes in NO$_2$ VCD in Asia include extensive decreases throughout Japan (-22%), South Korea (-39%),
Thailand (-7%), Pakistan (-14%) and Australia (-14%). Urban increases were observed in much of Central Asia, including
Turkmenistan (+21%), Kazakhstan (+22%) Mongolia (+75%).

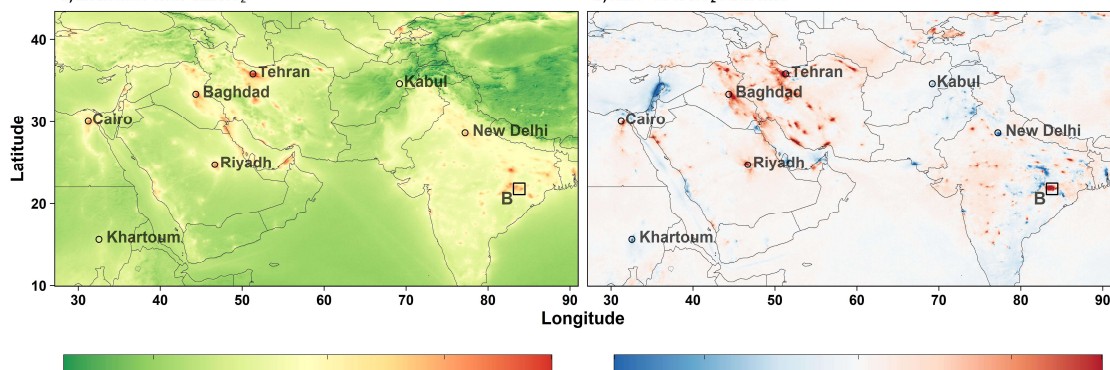

**Figure 4: Same as Fig. 2, but centered on the Middle East. Labeled black squares indicate the locations of mining regions highlighted**
**in Fig. 3.**
**3.2 Africa**
Johannesburg, South Africa and the surrounding region exhibited the largest NO$_2$ VCD for the African continent in 2024 (Fig.
S4). Numerous surface coal mines and coal-fired power plants, particularly to the east of Johannesburg, contribute to the
region's NO$_2$ signature (Shikwambana et al., 2020). Despite these elevated NO$_2$ levels, 2024 mean NO$_2$ VCDs in the city of
Johannesburg were 8% lower than in 2019. Northwest of Johannesburg in Limpopo Province, mining operations at the



Grootegeluk surface coal mine, together with two adjacent power plants (Faure et al., 2010; Shikwambana et al., 2020), produce
one of the largest $NO_2$ point sources in Africa, despite annual mean $NO_2$ VCDs at the site decreasing by 3.5 x $10^{15}$ molecules
$cm^{-2}$ from 2019 to 2024, or a decrease of 32% (Fig. 3d). The Cairo, Egypt urban region, in Northern Africa, represents the
second largest urban $NO_2$ signature in Africa in 2024. The 2024 annual mean $NO_2$ VCD in Cairo was 9.4 x $10^{15}$ molecules $cm^{-}$
$^2$, and elevated VCDs extend along the Nile River south of Cairo, as well as north into the Nile River Delta. Cairo exhibited
increased VCDs from 2019 to 2024 (+8%), as did regions immediately adjacent to the Nile River, while regions north into the
Nile River Delta exhibited decreased $NO_2$ VCDs.
In the Sudanese capital of Khartoum, $NO_2$ VCDs started decreasing in 2023, coinciding with the onset of conflict within Sudan
(Guo et al., 2023). This resulted in annual mean VCDs decreasing by 58% from 2019 to 2024 (Fig. S5). In a mining region
known as the Copperbelt in the south of the Democratic Republic of the Congo (DRC), broad $NO_2$ VCD increases were
observed, including at a large surface mine near Kolwezi. VCDs at the Kolwezi mine increased by 1.4 x $10^{15}$ molecules $cm^{-2}$
from 2019 to 2024, or an increase of 64% (Fig. 3e). Numerous surface mines exist in the region, with most observing increases
in NOx emissions from mining operations in recent years (Martínez-Alonso et al., 2023). Throughout the remainder of Africa,
moderate VCD enhancements were observed near most urban centers, with mean VCDs near most cities typically at or below
4 x $10^{15}$ molecules $cm^{-2}$ (Fig. 1a). Along the African Mediterranean coast, most urban areas showed increased $NO_2$ VCDs
through 2024. Other national capitals and major cities exhibited increased VCDs, including Abidjan, Ivory Coast (+41%);
Addis Ababa, Ethiopia (+34%); Kinshasa, DRC (+20%); and Dakar, Senegal (+15%).

## 3.3 Europe

$NO_2$ VCDs in Europe were largest in urban areas, with the largest 2024 mean VCD occurring in Moscow, Russia (15.5 x $10^{15}$
molecules $cm^{-2}$) (Fig. 5a). Broad enhanced 2024 annual mean VCDs exceeding 4 x $10^{15}$ molecules $cm^{-2}$ were observed in a
region encompassing Belgium, the Netherlands and western portions of Germany, with values exceeding 5 x $10^{15}$ molecules
$cm^{-2}$ in the Po River Valley of northern Italy.
From 2019 to 2024, decreases in $NO_2$ VCD occurred in 61% of all urban clusters in Europe. All cities with a population greater
than 1,000,000 experienced decreases, with the exception of Moscow (+29%) and other cities of western Russia, which
experienced increases (Fig. 5b). The broad decreases across large Europe cities are likely due to a combination of (1) continued
decreased emissions trends that accelerated during the COVID-19 pandemic, (2) continued transition to alternative energy
sources following the start of the Russia-Ukraine war in 2022 and (3) existing policies implemented within the EU (Matthias
et al., 2021; Rokicki et al., 2023; Cifuentes-Faura, 2022). These policies include the European Green Deal and European
Climate Law, which promote zero-emission vehicles, stricter vehicle emissions targets and updated industrial emissions
regulations.



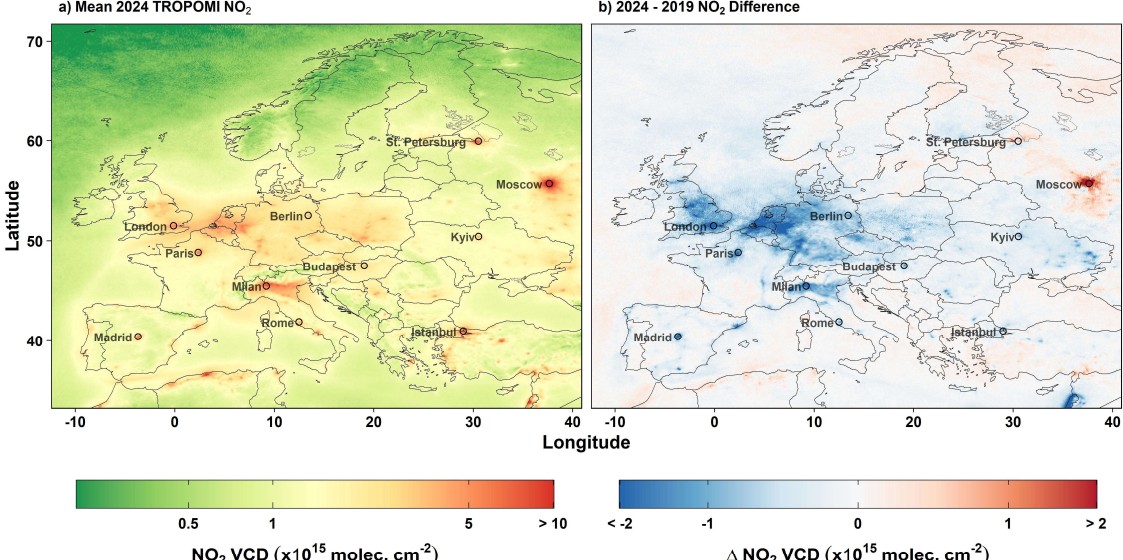


**Figure 5: Same as Fig. 2, but centered on Europe.**

**3.4 North America and South America**
Throughout North America, 2024 annual mean $NO_2$ VCDs were largest in urban regions, including Los Angeles (7.4 x $10^{15}$
molecules cm$^{-2}$), New York (7.0 x $10^{15}$ molecules cm$^{-2}$), Chicago (5.0 x $10^{15}$ molecules cm$^{-2}$), Mexico City (11.3 x $10^{15}$
molecules cm$^{-2}$) and Toronto (4.3 x $10^{15}$ molecules cm$^{-2}$), as well as near fossil fuel-fired power plant and mining operations
(Fig. 6a). A majority of cities in the U.S. and Canada exhibited decreased or unchanged $NO_2$ VCDs (Fig. 6b), with notable
exceptions being Phoenix, Arizona (+10%) and Dallas, Texas (+6%), which experienced increases (Fig. S6).
In Canada, the largest VCD decreases were observed in Alberta Province in and around Edmonton (-19%). In the U.S., aside
from decreases in urban environments, the largest changes were observed in remote areas near power plants, e.g. near the
decommissioned Navajo Generating Station in northern Arizona, the Four Corners Generating Station in northern New
Mexico, and the Hunter and Huntington Power Plants in central Utah (Goldberg et al., 2021). Oil, gas, and coal mining
operations influenced regional VCD changes as well, with annual mean $NO_2$ VCDs decreasing from 2019 to 2024 in the
Athabasca oil sands (+1%) in Northern Alberta (Fig. 3f), increases in the Permian (+29%) and Uintah (+35%) Basins in the
southwestern U.S. (Fig. 3g-h), and decreases in the Bakken (-16%) in North Dakota (Fig. 3i). Apparent within the U.S. is a
slight increase in background concentrations in rural regions, particularly in the Central and Western U.S. It is unclear if this
is due to an extension of the $NO_2$ lifetime due to decreasing VOCs and $O_3$ over this 6-year period (e.g., Laughner & Cohen
2019) or due to increased NOx emissions in rural areas or both. Further work should investigate this.



In Mexico, Central America and the Caribbean, the largest VCDs are observed near Mexico City (11.3 × $10^{15}$ molecules cm-
$^{2}$) and Monterrey, Mexico (7.7 × $10^{15}$ molecules cm$^{-2}$), with numerous other urban signatures. The largest increases were
observed at sites in Northern Mexico, including Mexicali (+31%) and Hermosillo (+32%) as well as a handful of regions with
decreased VCDs in northern Mexico, including Monterrey (-9%). VCDs also decreased near the capital city of Santo Domingo,
Dominican Republic (-28%), and increased near Havana, Cuba (+39%).
In South America, the largest VCDs are observed near Lima, Peru (6.3 × $10^{15}$ molecules cm$^{-2}$); Santiago, Chile (9.7 × $10^{15}$
molecules cm$^{-2}$); and Sao Paulo, Brazil (7.3 × $10^{15}$ molecules cm$^{-2}$) (Fig. S7). Regions near Santiago experienced some of the
largest decreases in VCD in South America (-19%), while Quito, Ecuador experienced the largest increase (+86%).

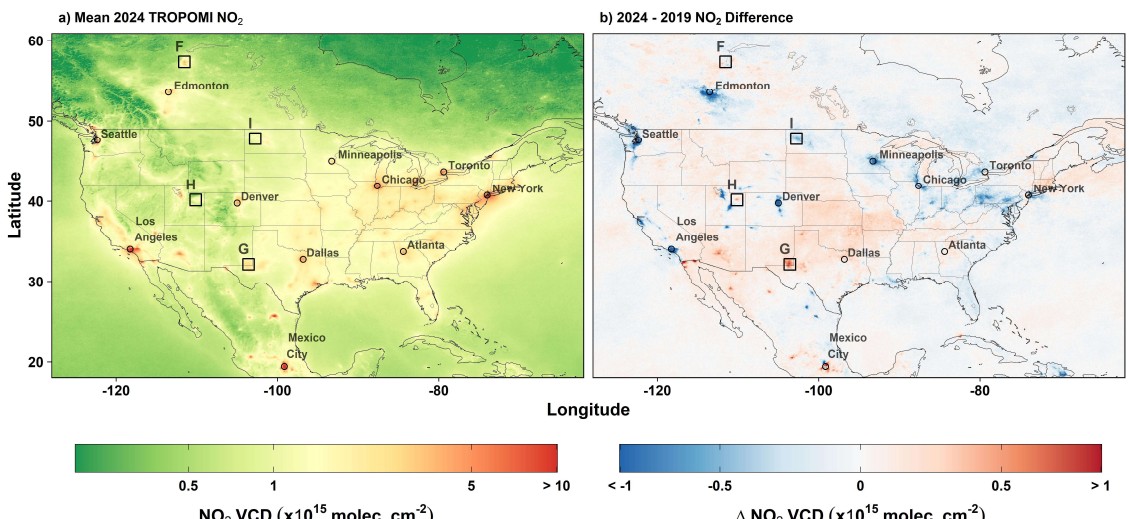


**Figure 6: Same as Fig. 2, but for North America. Squares and numbers represent select oil and gas regions highlighted in Fig. 3.**

**4 Urban-level NO$_2$ VCD trends**
Using the method outlined in Section 2.2.1, the GHS-SMOD urban cluster boundaries are used to determine mean TROPOMI
NO$_2$ concentrations for all urban clusters globally with a minimum population of 50,000. Looking at VCD changes from 2019
to 2024 in the 50 cities representing the ten most populous urban clusters on each continent, with Asia and Oceania considered
jointly, East Asian cities represent four and European cities represent five of the ten largest VCD decreases (Fig. 7a). Seoul
experienced the greatest reduction in NO$_2$ VCD of any of these 50 cities, with annual average levels from 2019 to 2024
decreasing by 7.4 × $10^{15}$ molecules cm$^{-2}$ (Fig. 7b), or nearly -40% (Fig. 7c). London, England produced the greatest NO$_2$ VCD





decrease of the ten most populous European cities, with a mean decrease of $2.5 \times 10^{15}$ molecules cm$^{-2}$ (Fig. 7b), or -34%. This
decrease occurred alongside the introduction of the city's ultra-low emission zone introduced in 2019 and expanded in 2023,
which has been shown to decrease local NO$_2$ concentrations (Hajmohammadi and Heydecker, 2022).
Large South American cities generally experienced minimal changes in NO$_2$ VCD, with relative changes typically less than
±5% (Fig. 7c). The most notable exception is Santiago, Chile, which experienced a mean VCD decrease of nearly 20% from
2019 to 2024. The largest North American cities mostly experienced moderate VCD decreases, with the largest absolute
decreases occurring in Los Angeles (-13.7%) and the San Francisco Bay Area (-20.4%), and largest increases occurring in the
Mexican cities of Guadalajara (+9.9%) and Mexico City (+4.6%). Chicago and New York City, two of the largest cities in the
U.S., also experienced decreases, though less pronounced.

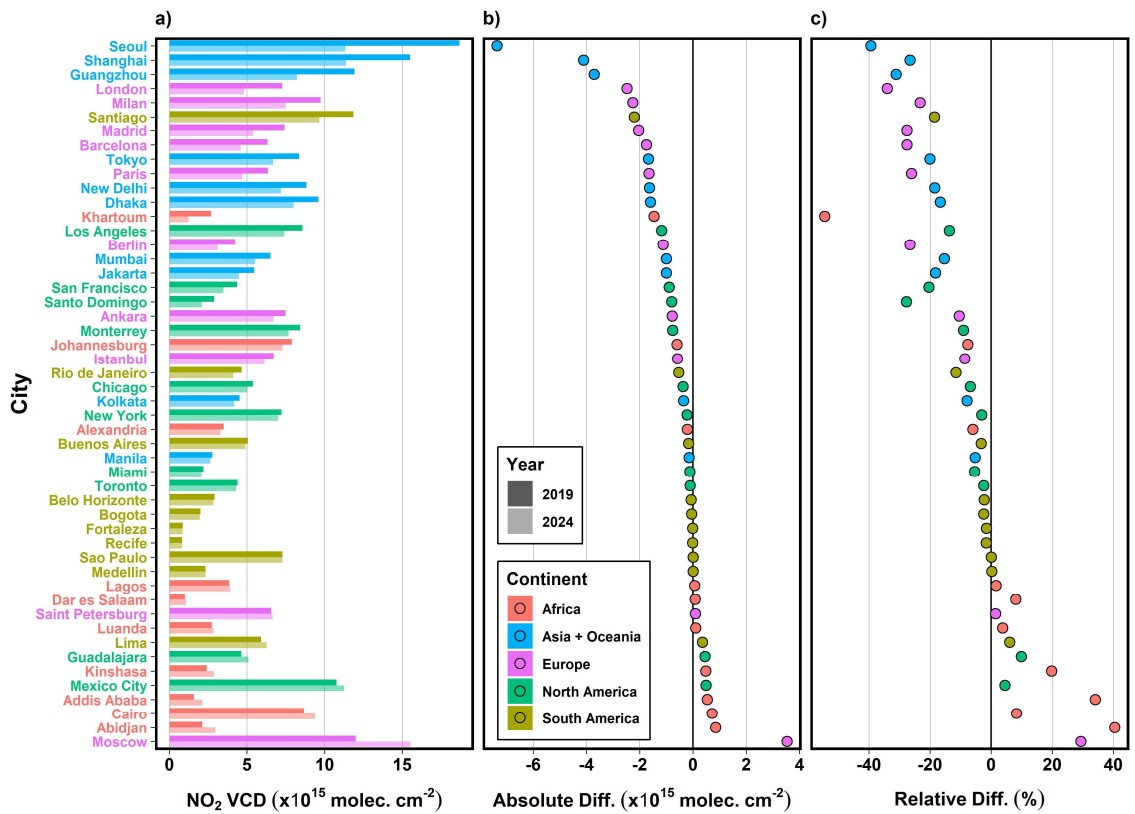


**Figure 7: (a) NO$_2$ VCD in 2019 (dark bars) and 2024 (light bars) for the 10 most populous urban clusters on each continent, based**
**on GHS-SMOD populations. (b) Absolute difference in NO$_2$ VCD for each city from 2019 to 2024. (c) Relative percent change in**
**VCD from 2019 to 2024. Colors correspond to the respective continent for each city. Cities are ordered by magnitude of absolute**
**VCD decrease.**



Most of the largest African cities experienced increased $NO_2$ VCDs from 2019 to 2024, with Abidjan, Ivory Coast experiencing
the largest urban increase of $0.85 \times 10^{15}$ molecules $cm^{-2}$ (Fig. 7b), or an increase of 40.5% (Fig. 7c). Additional notable African
increases are Cairo, Egypt (+8.3%), Addis Ababa, Ethiopia (+34.1%) and Kinshasa, DR Congo (+19.9%). The largest decrease
on the African continent was observed in the Sudanese capital of Khartoum, which experienced an average decrease of $1.46 \times$
$10^{15}$ molecules $cm^{-2}$ (Fig. 7b) or a decrease of 54.5% (Fig. 7c). These strong VCD decreases in Khartoum coincide with conflict
in the country, causing large portions of the city to be displaced, impacting $NO_2$ concentrations (see Sec. 3.2).
Of the cities presented in Fig. 7, the three largest absolute decreases between 2019 and 2024 were in the East Asian cities of
Seoul, South Korea (Fig. 8a), Shanghai, China (Fig. 8b) and Guangzhou, China (Fig. 8c). Seoul experienced decreases greater
than $7 \times 10^{15}$ molecules $cm^{-2}$ from 2019 to 2024, largely due to effective policies implemented by the South Korean government
since the early 2000s to reduce local emissions, as well as trends in emissions that began following the COVID-19 pandemic
(Ho et al., 2021; Seo et al. 2021). The observed annual decreases in these East Asian cities were primarily driven by decreases
during the winter months (Fig. 7d). European cities also experienced some of the largest decreases in VCD, with the three
largest decreases occurring in London, UK (-34%); Milan, Italy (-23%); and Madrid, Spain (-28%) (Fig. S8). Three cities with
notable increases include Moscow, Russia (+29%), Baghdad, Iraq (+17%) and Riyadh, Saudi Arabia (+13%) (Fig. S9).
Moscow experienced the largest $NO_2$ VCD increase of any GHS-SMOD city through 2024, with a mean increase of $3.5 \times 10^{15}$
molecules $cm^{-2}$ (Fig. 7b). The increasing trend in Moscow accelerated in early 2022 (Fig. S9), following the onset of the
Russia-Ukraine war in Ukraine, when monthly mean $NO_2$ VCDs for March reached $59 \times 10^{15}$ molecules $cm^{-2}$ (see Sec. 3.3).

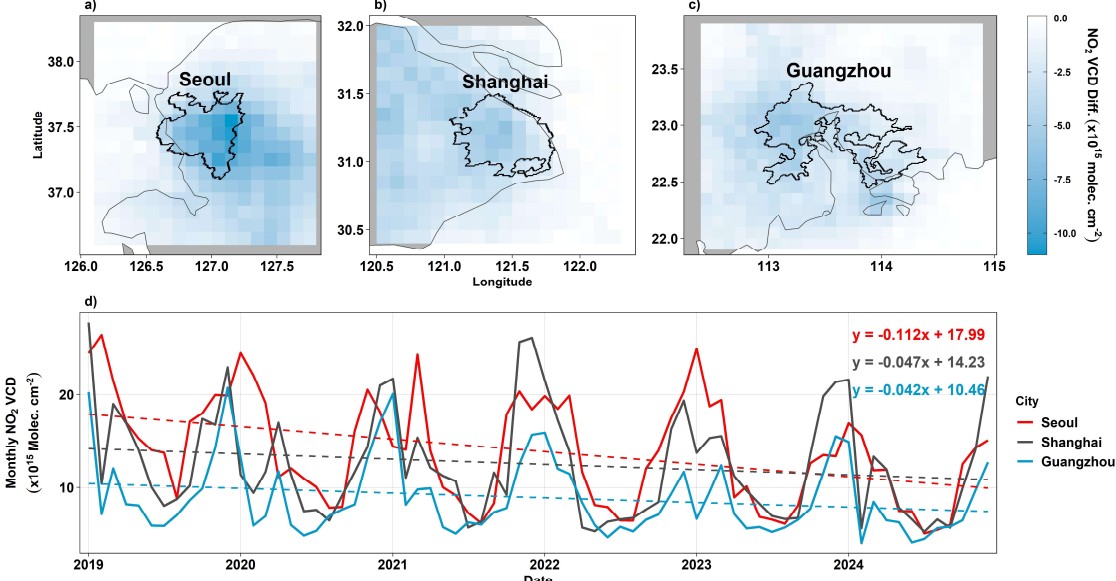


**Figure 8: Absolute change in mean annual NO₂ VCD from 2019 to 2024 for three East Asian cities: (a) Seoul, South Korea, (b) Shanghai, China and (c) Guangzhou, China. Colors in panels a-c show magnitude of VCD change, thin lines show national borders or coastlines, and thick lines show the GHS-SMOD urban boundary. (d) Solid lines show monthly mean TROPOMI NO₂ VCD from 01/2019 through 12/2024, colored by city. Dashed lines and equations show ordinary least-squares regression trends, with the slope representing the change in NO₂ VCD per month, and the y-intercept representing the intercept for January, 2019.**

## 5 Aggregated trends in urban TROPOMI NO₂

TROPOMI NO₂ changes from 2019 to 2024 for larger urban clusters (i.e. clusters with a population greater than 500,000) are shown in Fig. 9. This represents approximately 1000 of the most populated urban clusters, or just over 9% of all urban clusters in the GHS-SMOD dataset. 15.2% of these cities are in Africa, 57.8% are in Asia and Oceania, 10.9% are in Europe, 9.1% are in North America and 6.9% are in South America. On average, annual mean NO₂ VCDs are characterized by a decrease of about 10% from 2019 to 2020; previous work has attributed such decreases to the COVID-19 pandemic (Cooper et al., 2022). Africa was the exception to these 2020 decreases (Fig. 9a), which saw average VCDs largely unchanged for that year. Through 2024, African cities experienced a gradual increase in VCD, with larger cities exhibiting a larger fractional increase in NO₂ (Fig. 9a). The largest percent increase occurred in Abidjan, the capital city of Ivory Coast, which experienced an increase in NO₂ VCD of more than 40% from 2019 through 2024. Khartoum, Sudan experienced the largest percent decrease of any large African City, with mean VCDs decreasing by nearly 60% through 2024, with most of that decrease accelerating in the Spring of 2023 (Fig. S5). This decrease coincides with the onset of conflict within Sudan, which heavily impacted the capital city of Khartoum, leading to the displacement of much of the Khartoum population outside the city (Guo et al., 2023).



332

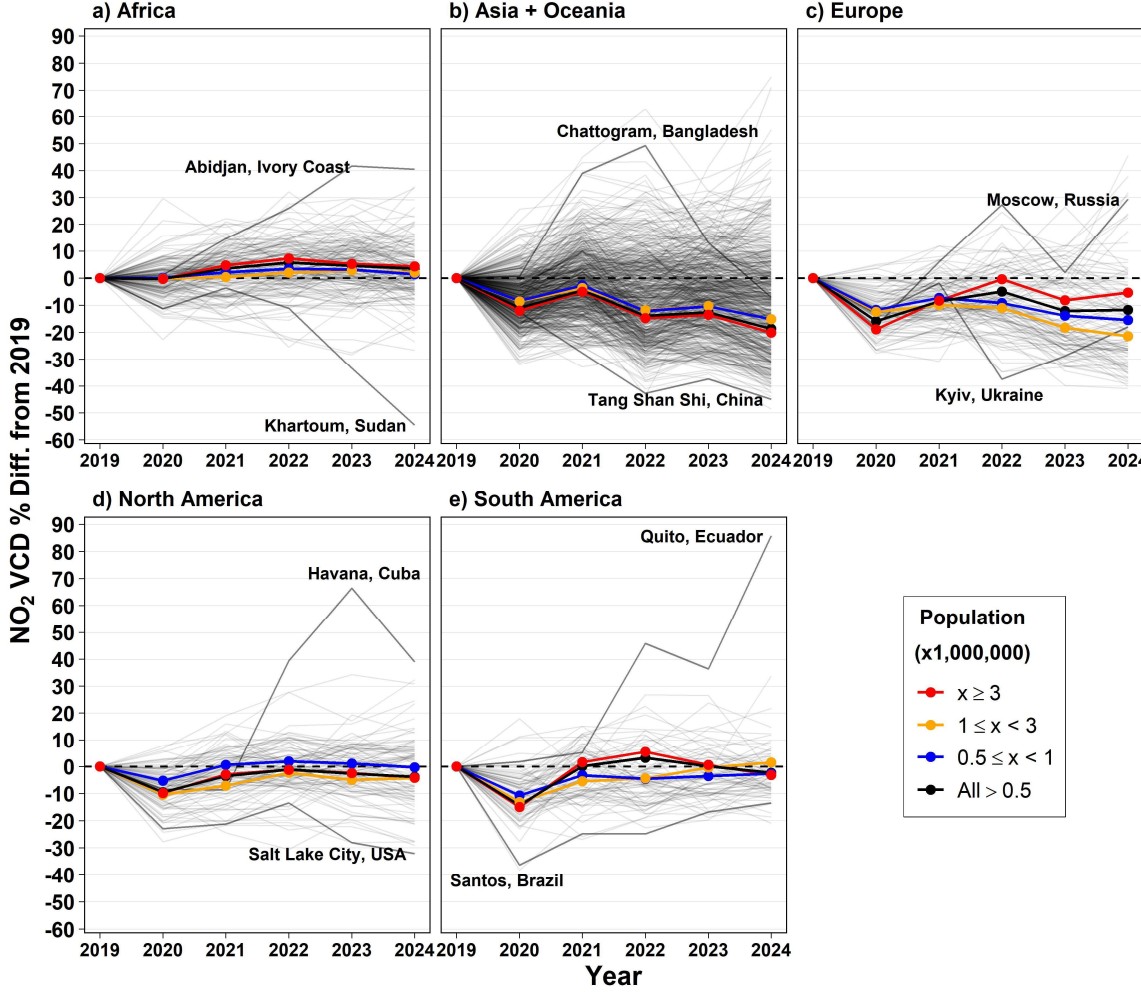

333

**Figure 9: Percent change in annual mean TROPOMI tropospheric NO₂ vertical column densities (VCD) for individual GHS-SMOD urban clusters with a population of at least 500,000 (gray lines), relative to 2019 values. Population-weighted percent change is shown for urban clusters with a population between 500,000 and 1 million (blue), between one and three million (yellow), greater than three million (red) and all clusters with a population greater than 500,000 (black). Results are separated by continent for (a) Africa, (b) Asia and Oceania, (c) Europe, (d) North America and (e) South America.**

Asian cities, representing a majority of all urban clusters globally, experienced an average population-weighted NO₂ VCD decrease of approximately 19% from 2019 to 2024 (Fig. 9b). Urban clusters with a population greater than 3 million experienced the largest decreases, with mean VCDs in those cities decreasing by 20% through 2024, with the majority of these



decreases occurring in Chinese cities. One notable decrease in Asia occurred in the Chinese city of Tangshan Shi, located to
the east of the Chinese capital of Beijing, which experienced an $NO_2$ VCD decrease of nearly 45% from 2019 to 2024. The
largest increase in Asia through 2024 occurred in the Mongolian capital of Ulaanbaatar, where VCDs have increased by more
than 70%. Numerous Bangladeshi cities, including Chattogram, experienced substantially increased VCDs from 2020 through
2022, with VCDs decreasing again by 2024 to the near 2019 levels (Fig. S10). Tehran, Iran by far has the largest annual average
VCD in the TROPOMI tropospheric $NO_2$ record for all GHS-SMOD cities, with annual mean values remaining above 30 ×
$10^{15}$ molecules $cm^{-2}$ throughout the entirety of the TROPOMI record (Fig. S11).
The impact of the COVID-19 pandemic on $NO_2$ VCDs is particularly stark in Asia, due to the multiple waves of COVID-19
related lockdowns and closures in China, leading to reduced $NO_2$ levels. Initial lockdowns in 2020 led to widespread VCD
decreases in China, which were followed by a rebound in levels in 2021 (Fig. S12). A resurgence of the virus in 2022 led to
multiple further lockdowns throughout the year, some lasting for months (Zheng et al., 2021; Zhao et al., 2024), that ultimately
resulted in reduced VCDs. Chinese cities continued to experience decreased $NO_2$ VCDs in 2023 and 2024, in large part due to
effective emissions reduction policies (Li et al., 2024).
Column $NO_2$ in European cities experienced the most pronounced decrease in column $NO_2$ of any continent in 2020, with
larger cities with a population greater than three million experiencing a nearly 20% reduction in population-weighted VCD
(Fig. 9c). $NO_2$ VCDs rebounded marginally in 2021 and 2022, followed by decreases into 2023 and 2024, although decreases
are more pronounced when only analyzing cities in the 27 member countries of the European Union, as of 2024 (Fig. S13).
One notable feature within the European annual average VCDs are the contrasting VCD trends in Russian and Ukrainian cities
in 2022, at the onset of the Russia-Ukraine War (Fig. S14). In the Ukrainian capital of Kyiv, annual VCDs dropped nearly
40% in 2022 relative to 2019, coinciding with a large portion of the city fleeing due to conflict in and near the city. To contrast
this, VCDs increased nearly 30% in the Russian capital of Moscow during the same period. Following 2022, VCDs in Kyiv
increased steadily, while in Moscow, concentrations decreased in 2023 then increased sharply again in 2024.
In North America, most cities experienced a decrease in annual $NO_2$ VCD of less than 10% in 2020, with concentrations
generally rebounding to 2019 levels by 2024 (Fig. 9d). Havana, Cuba was a notable exception of North American cities, with
VCDs increasing by nearly 70% through 2023 relative to 2019, with a slight decrease in 2024. Cities in the western U.S., such
as Salt Lake City and Denver experienced some of the largest percent decreases on the continent, decreasing by approximately
30% through 2024. In South America, most cities experienced a 10% VCD decrease in 2020, with mean concentrations
rebounding to 2019 values by 2021 and remaining around those levels through 2024 (Fig. 9e). One notable exception is Quito,
Ecuador, which experienced a VCD increase of over 85% through 2024. Santos, Brazil, an active port town southeast of São
Paulo, experienced one of the largest VCD decreases in South America, with a 35% decrease in VCDs from 2019 to 2020,
followed by sustained, gradual annual increases through 2024.



Aggregating the NO$_2$ VCD changes to the country level and taking into account the population of each urban cluster (Eq. 1),
the urban and population-weighted NO$_2$ VCD decreases for countries with an urban population of at least nine million were
largest in the East Asia, including South Korea, China and Japan, as well as countries of Western and Central Europe (Fig.
10). Urban population-weighted VCD decreases in South Korea were particularly pronounced, with concentrations decreasing
by $5.6 \times 10^{15}$ molecules cm$^{-2}$ from 2019 to 2024. Germany experienced the largest VCD decrease in Europe through 2024,
with a decrease of $1.7 \times 10^{15}$ molecules cm$^{-2}$. Chile saw the largest decrease in urban NO$_2$ VCD of any South American country,
due in large part to VCD decreases in the capital city of Santiago.

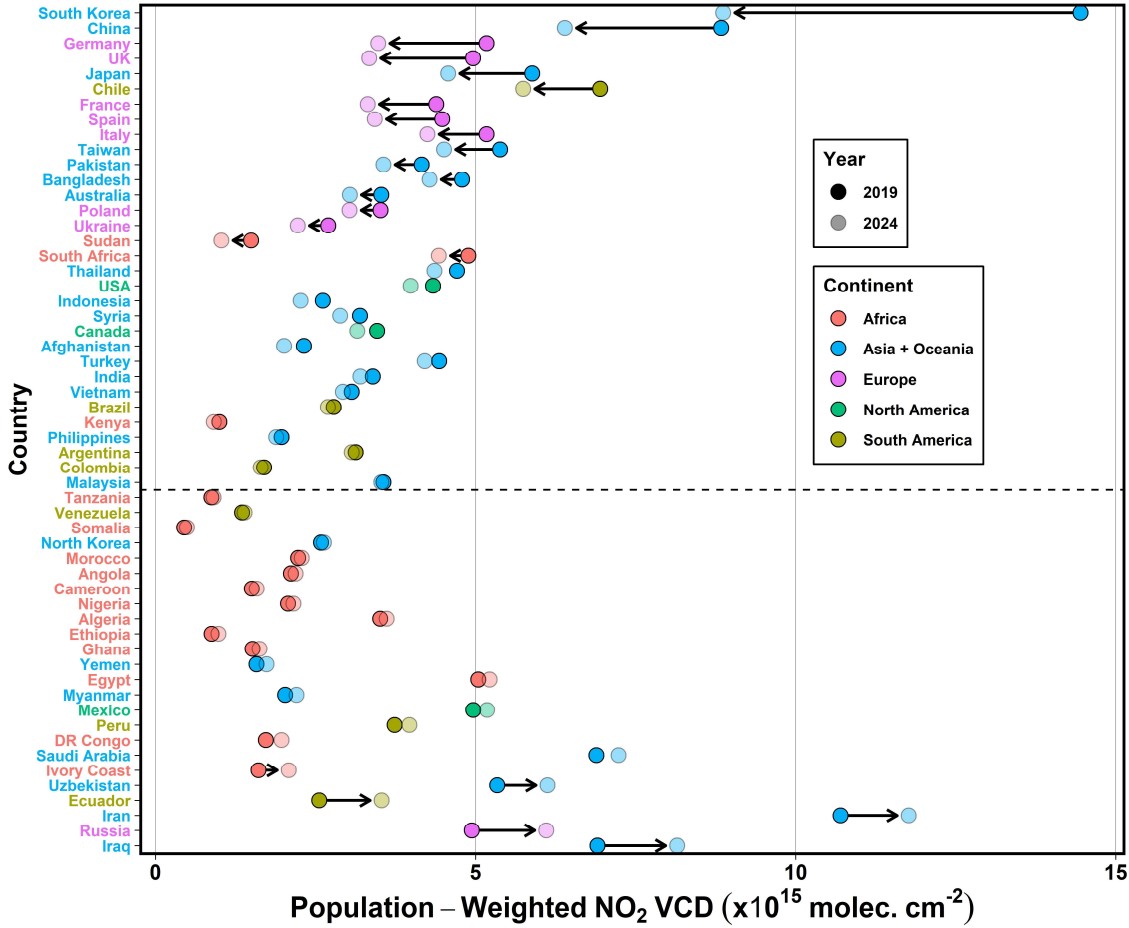


**Figure 10: Urban population-weighted NO$_2$ VCD changes from 2019 to 2024 for the 56 countries with an urban population of at**
**least nine million, based on urban cluster populations provided from GHS-SMOD. Countries are ordered by the magnitude of VCD**
**decrease and colored by continent. Darker points represent 2019 VCDs and lighter points represent 2024 VCDs. Arrows indicate**





**the direction of VCD change from 2019 to 2024. The dashed line separates countries that experienced population-weighted VCD decreases (above line) and increases (below line).**

Of countries with increased VCDs and an urban population greater than nine million, about half were in Africa. The majority of increases in these African countries were relatively minor, with country-level urban VCDs typically increasing by less than $0.25 \times 10^{15}$ molecules cm$^{-2}$ through 2024 (Fig. 10). Middle Eastern and Central Asian countries experienced some of the largest urban VCD increases, with Iraq experiencing the largest increase of any larger country ($+1.2 \times 10^{15}$ molecules cm$^{-2}$). Of the most-populous European countries, Russia was the only country to experience increased population-weighted NO$_2$ VCDs through 2024 ($+1.16 \times 10^{15}$ molecules cm$^{-2}$).

We further identify notable changes in countries with a GHS-SMOD urban population less than nine million and therefore excluded from Fig. 10. Less-populated countries in Africa saw either increasing or little-changed NO$_2$ VCDs from 2019 to 2024 (Fig. 11). In Asia, Mongolia experienced an increase of $2.05 \times 10^{15}$ molecules cm$^{-2}$ through 2024, or an increase of 75%, the largest population-weighted percent increase of any Asian country. We note that Mongolia has just three GHS-SMOD urban clusters, two of which are located in or near the capital city of Ulaanbaatar, where the bulk of the country-level increases were observed. In Sri Lanka, VCDs decreased by $0.5 \times 10^{15}$ molecules cm$^{-2}$, or a decrease of 27%, one of the larger decreases for Asian countries. In Europe, both Belgium and the Netherlands experienced VCD decreases through 2024 that exceed 30%.

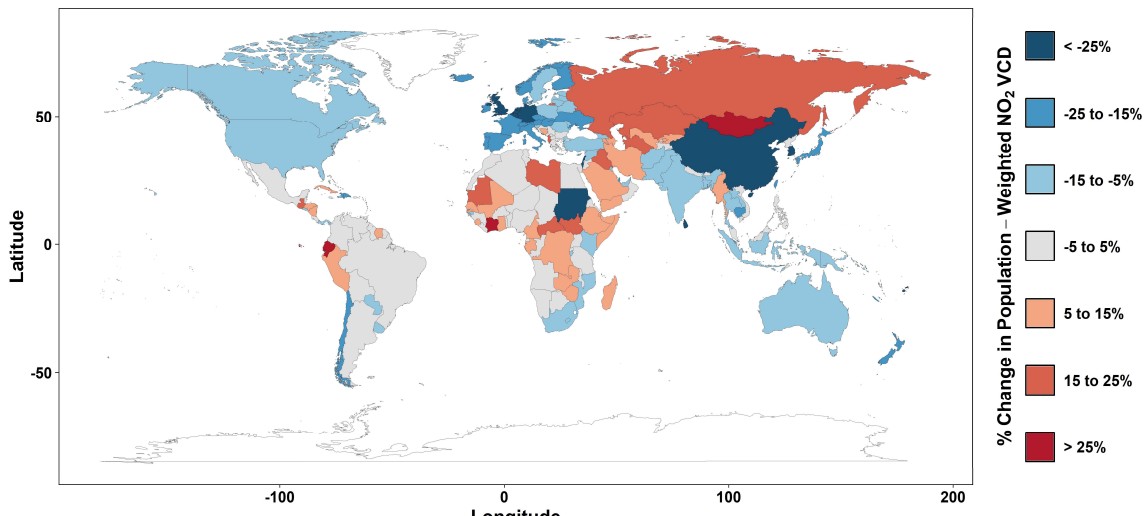

**Figure 11: Spatial representation of the percent change in urban population-weighted NO₂ VCD for all countries globally from 2019 to 2024, binned by the magnitude of percent change. Increases greater than 5% are shown in reds, decreases less than 5% in blues, and countries with a change between -5% and 5% in gray.**



## 6 Influence of background NO₂ and seasonal variability on urban NO₂

Urban $NO_2$ concentrations are not only influenced by local emissions, but also by advection of upwind pollutants into the urban boundary. We account for the role that upwind background concentrations may play in urban $NO_2$ concentrations by identifying changes in the urban enhancement of $NO_2$ ($VCD_{ENH}$), represented by the difference between $NO_2$ VCDs in the urban cluster and the background. By removing the background concentrations, $VCD_{ENH}$ more closely represents the portion of the urban VCD that is primarily a result of local, urban emission sources.

In Africa, moderate-sized cities with a population between one and three million experienced the smallest relative increase in $VCD_{ENH}$ through 2024 (+4.9%), while smaller and larger cities experienced larger increases (Fig. 12a). Notably, African cities on average did not experience decreased $VCD_{ENH}$ in 2020 at the onset of the COVID-19 pandemic, a distinct feature observed on all other continents. Regardless of population size, African cities experienced an average $VCD_{ENH}$ increase of +6% through 2024. In Asia and Oceania, cities experienced sustained decreases in $VCD_{ENH}$ regardless of the city population, with a mean decrease of -22.7%, although larger cities experienced more pronounced decreases (Fig. 12b). In contrast, changes in $VCD_{ENH}$ in European cities largely depended on the population of the city, with smaller (-17.6%) and moderate-sized (-26.7%) cities exhibiting the largest decreases, while larger cities (-5.3%) experienced lesser decreases on average (Fig. 12c). In North America, smaller cities between 500,000 and 1 million saw a -8% decrease in $VCD_{ENH}$ in 2020, but quickly rebounded in 2021 to near 2019 levels, which were sustained through 2024 (Fig. 12d). Moderate and large North American cities also rebounded following the dip in 2020, however $VCD_{ENH}$ remained approximately 7.5% below 2019 levels by 2024. In South America, cities experienced a $VCD_{ENH}$ decrease of 16% on average in 2020, with concentrations in cities of all populations rebounding to near 2019 levels by 2024 (Fig. 12e).





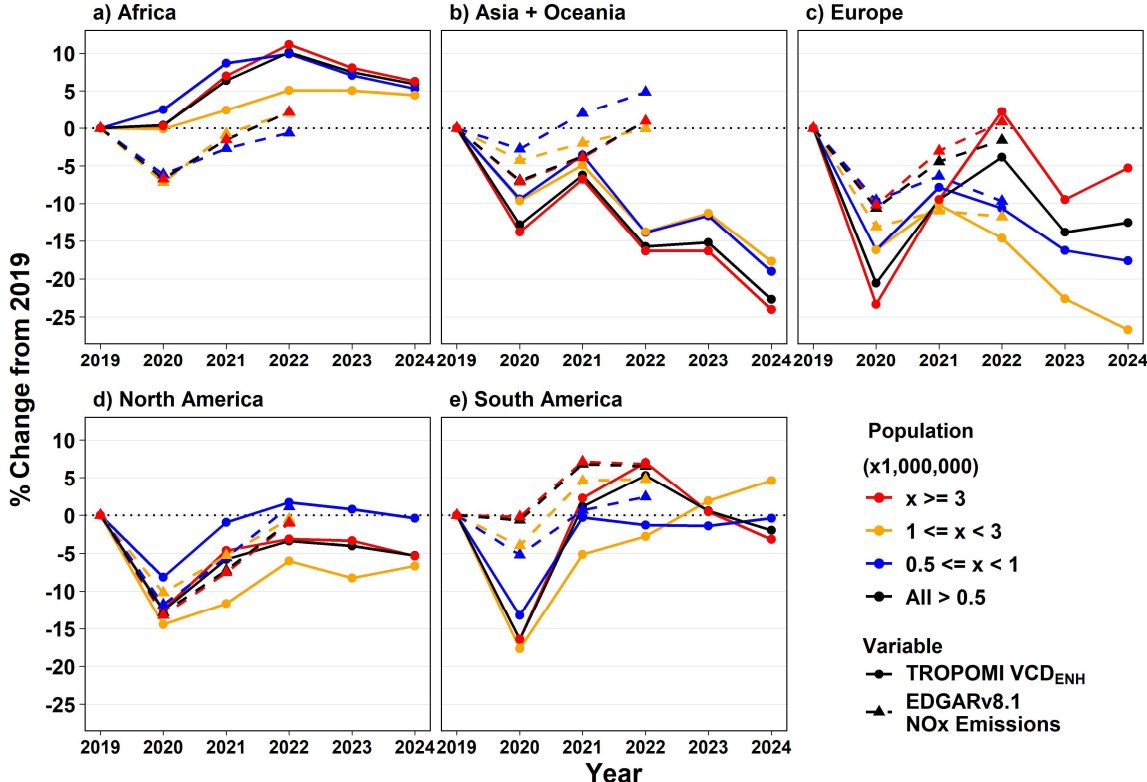


**Figure 12: Percent change in population-weighted TROPOMI NO₂ VCD urban enhancements (solid lines; 2019 - 2024) and EDGARv8.1 NOx emissions (dashed lines; 2019 - 2022) for GHS-SMOD urban clusters, relative to 2019 levels. Colors represent urban cluster population range, and results are separated by continent (a-e).**


Assuming that the percent change in VCD$_{ENH}$ relative to a baseline year can be attributed to changes in NOx emissions within
each urban cluster, we additionally evaluate changes in EDGAR NOx emissions averaged for each continent, with emissions
estimates available through 2022. In African cities, a mean difference of -7.7% was exhibited between the percent changes in
EDGAR NOx emissions and VCD$_{ENH}$ relative to 2019, indicating a potential underestimate in EDGAR NOx emissions for this
period (Fig. 12a). Cities in Asia and Oceania experienced VCD$_{ENH}$ that tracked relatively well with EDGAR NOx emissions
from 2019 to 2021, with a mean difference of +4.2% between emissions and VCD$_{ENH}$. However, emissions showed further
increases in 2022, while VCD$_{ENH}$ exhibited a sharp decrease for that year. This resulted in a percent difference of +16.7%
between emissions and VCD$_{ENH}$ in 2022 relative to 2019 levels (Fig. 12b). The 2022 VCD$_{ENH}$ decrease coincided with large
lockdowns in China related to the COVID-19 pandemic, suggesting that EDGAR emissions may not reflect emissions



decreases during that lockdown period. In Europe and North America, EDGAR NOx emissions and VCD$_{ENH}$ exhibited a
similar change relative to 2019 levels through 2022, with a mean difference of +5.7% and +0.2%, respectively, suggesting
more accurate EDGAR NOx emissions for cities on those continents (Fig. 12c,d). In South America, the mean percent change
relative to 2019 was +7.6% higher for EDGAR NOx emissions than VCD$_{ENH}$ (Fig. 12e); however, EDGAR emissions do
correlate with changes in VCD$_{ENH}$, e.g. a flat or slower increase from 2021 to 2022 for both VCD$_{ENH}$ and EDGAR. The better
agreement in Europe and North America than other continents could be due to a higher availability of observational constraints
on emissions, leading to more accurate changes in emissions from year to year.
To identify the impact that different seasons may have on annual trends, we evaluate changes in urban population-weighted
NO$_2$ VCDs for May – September and November – March. In African cities (Fig. 13a), mean VCDs increased by $0.1 \times 10^{15}$
molecules cm$^{-2}$ during November – March through 2024, with little to no change occurring on average during May –
September. In Asian (Fig. 13b) and North American cities (Fig. 13d), the bulk of the observed annual decreases through 2024
occurred during the winter months, with average winter decreases of $-1.8 \times 10^{15}$ molecules cm$^{-2}$ and $-0.5 \times 10^{15}$ molecules cm$^{-}$
$^{2}$, respectively. Despite the generally larger absolute changes during winter months in Asia and Oceania, the relative percent
changes for the summer and winter months exhibited more similar behavior (Fig. S15). In European cities, population-weighted
VCDs decreased by $-0.4 \times 10^{15}$ molecules cm$^{-2}$ (-10%) through 2024 during the summer months, while winter month changes
remained negligible, despite a sharp increase in winter-time levels in 2022 during the onset of the Russia-Ukraine war (Fig.
13c). Seasonal changes impacted South American cities less than cities on other continents through 2024 (Fig. 13e), with mean
winter and summer VCDs both changing by less than $0.3 \times 10^{15}$ molecules cm$^{-2}$ through 2024.



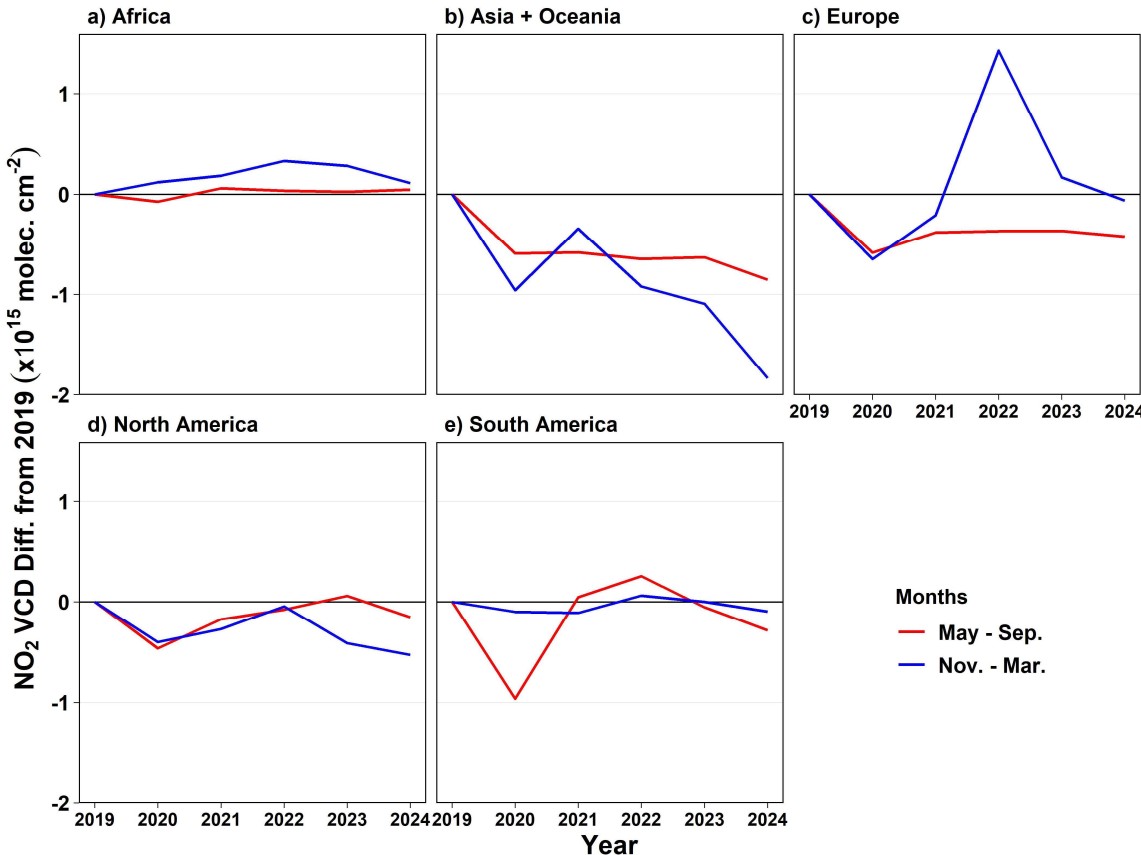

**Figure 13: Change in population-weighted urban NO₂ VCDs averaged for GHS-SMOD urban clusters from 2019 to 2024 for May – September (red lines) and November – March (blue lines) in (a) Africa, (b) Asia and Oceania, (c) Europe, (d) North America and (e) South America.**

**7 Conclusions**

We present a global analysis of urban TROPOMI tropospheric NO₂ trends from 2019 to 2024 using GHS-SMOD-defined urban boundaries, encompassing more than 11,500 cities. Our results reveal widespread decreases in NO₂ across cities in Asia and Oceania (-17% on average), Europe (-17%), and North America (-5%), with particularly strong reductions in cities including Seoul (-40%), Guangzhou (-30%), and London, England (-34%). These decreases generally reflect a combination of long-term emissions control policies and economic incentives, indicating polices to tackle NO₂ pollution have broadly worked. COVID-19 induced reductions in activity often caused a temporary NO₂ reduction but is unlikely to have caused much of the long-term changes between 2019 and 2024. Conversely, urban NO₂ in Africa has gradually increased over the same period,



with Abidjan (+40%) and Addis Ababa (+35%) leading the continent's upward trend. With the exception of May-September
in 2020, South America exhibited little mean VCD change from 2019 to 2024, with being Santiago (-19%) being a notable
exception. Population-weighted $NO_2$ VCDs increased in countries in the Middle East and much of Africa, highlighting a
potential degradation in air quality in regions of the world that lack extensive ground-level monitoring.
Evaluating annual changes in TROPOMI $NO_2$ urban enhancements ($VCD_{ENH}$)—the difference between mean urban and
background VCDs—against changes in EDGAR NOx emissions, we show that changes in $VCD_{ENH}$ scales best with changes
in EDGAR NOx in European and North American cities, with mean percent differences of +5.7% and +0.2% relative to 2019
levels, respectively, and scale worse in other parts of the globe, revealing potential discrepancies in emissions inventories. This
mismatch is particularly evident in African (-7.7%) and Asian (+8.3%) cities, and may stem from rapidly evolving emission
sources or limitations in the EDGAR bottom-up inventory methods. Similar discrepancies in emissions inventories in the
Global South have been reported in previous studies (Ahn et al., 2023), suggesting a systematic emissions underestimation in
regions where unmonitored emissions activity may be significant.
In most regions, VCD trends from 2019 to 2024 were driven by changes during the colder months (November – March). This
was most pronounced in Asian cities, where mean cold season VCDs decreased by $-1.2 \times 10^{15}$ molecules $cm^{-2}$ (-18%) on from
2019 to 2024, compared with warm season VCD decreases of $-0.5 \times 10^{15}$ molecules $cm^{-2}$ (-13%). Large changes in $NO_2$ were
not confined to urban regions alone. We identified localized increases near fossil fuel and other mining operations, including
in the Santanghu Basin in China (+172%), the Permian (+19%) and Uintah (+35%) Basins in the U.S., and the Copperbelt
region of the DRC (+64%), signaling expanding industrial activity. In Khartoum and Kyiv, conflict and displacement drove
sharp reductions in $NO_2$, demonstrating the utility of satellite data in detecting societal disruptions.
Several limitations of this work should be noted. First, satellite $NO_2$ column densities may not always reflect surface-level $NO_2$
concentrations, particularly in regions with vertically elevated sources. In urban areas dominated by surface-based
transportation emissions, $NO_2$ VCDs are likely more representative of surface exposure. However, in areas with tall-stack
sources, such as power plants, $NO_2$ columns may be decoupled from near-surface levels. Second, we assume static city
boundaries defined by the 2023 version of GHS-SMOD, with population estimates from 2020. This is likely a reasonable
approximation for urbanized regions in Europe and North America, where built-up area changes are slow, but may introduce
uncertainty in rapidly urbanizing regions of Africa and Asia over a six-year period. Future analyses could incorporate time-
varying urban boundaries to address this.
Taken together, these results demonstrate the utility of high-resolution satellite instruments for characterizing both broad
regional trends and localized pollution changes, and linking with anthropogenically induced factors such as urban growth,
industrial expansion, policy interventions, and conflict. This highlights potential in using TROPOMI observations as an
accountability agent to determine how local changes in human activities affect local and global air pollution. As the TROPOMI
record lengthens and newer, geostationary satellites come online and begin to detect changes in atmospheric composition,



continued space-based monitoring will be essential for improving our understanding of atmospheric composition and chemistry
around the globe.

**Data Availability.**

The level 3 annual and monthly average TROPOMI $NO_2$ VCDs are available at 10.5067/ACADNS5UBWPQ and
https://doi.org/10.5067/KKPPL39PEIGE, respectively. The GHS-SMOD urban boundaries can be downloaded from
https://human-settlement.emergency.copernicus.eu/download.php?ds=smod. The EDGARDv8.1 NOx emissions can be
downloaded from https://edgar.jrc.ec.europa.eu/dataset_ap81.

**Supplement.**

The supplement contains additional figures related to the study, including: S1 All GHS-SMOD urban clusters. S2 Data
disaggregation example. S3 Satellite view of surface mines. S4 Spatial plot of African $NO_2$. S5 Khartoum $NO_2$ time series. S6
$NO_2$ increases in three U.S. cities. S7 Spatial plot of South American $NO_2$. S8 Annual mean $NO_2$ VCDs for Bangladeshi cities.
S9 Annual mean $NO_2$ in Iran. S10 Annual mean VCDs in Chinese cities. S11 Annual mean $NO_2$ changes in the European
Union. S12 Annual mean $NO_2$ changes in Russian and Ukrainian cities. S13 Seasonal $NO_2$ changes by continent.

**Author Contribution.**

D.H. and D.G. contributed to the project design. D.G. processed and provided the annually- and monthly-averaged $NO_2$ vertical
column densities. All authors edited the manuscript.

**Competing Interests.**

The authors declare that they have no conflict of interest.

**Acknowledgements.**

This work was supported by National Aeronautics and Space Administration (NASA) Health and Air Quality Applied Sciences
Team (HAQAST) grant #80NSSC21K0511 and NASA Aura Atmospheric Composition Modeling and Analysis Program
(ACMAP) grant #80NSSC23K1002.



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
