# Peer review of "TROPOMI NO2 for urban and polluted areas globally from 2019 to 2 2024"

_EGUsphere, 2025_

## Author Comment (AC1)

Huber and co-workers present a global scale analysis of $NO_2$ trends over large cities based on satellite TROPOMI column data. The study covers 6 years of observations, from 2019 to 2024, and includes the $NO_2$ sudden drop and subsequent rebound in many cities due to the COVID-19 disruptions, as well as air pollution changes in response to policy regulations, economic growth, and armed conflicts. Based on population-weighted annual columns, cities of Asia, Europe and North America show a $NO_2$ decline, whereas African cities show a moderate increase. Although not strictly comparable, the observed trends and the anthropogenic $NOx$ emission trends from the EDGARv8.1 bottom-up inventory give an indication that emission trends are underestimated in inventories, in particular in the Southern Hemisphere. The manuscript is well written but too lengthy, and the figures seem to support the drawn conclusions.

Thank you for this synopsis, and for taking the time to review our manuscript. We have made substantial changes to both the content and structure of the revised manuscript, and we feel that these changes greatly improve the clarity and quality of our work.

Although I appreciate the importance of the topic, there are several important issues with the paper. First, the trend analysis lacks an estimate of the statistical significance. I have serious doubts that the calculated trends in most cities are different from zero given the uncertainty in the satellite measurement and the limited number of available observations per city. The findings are therefore not sound. It is crucial to include a robust uncertainty analysis, and I argue that part of the conclusions will considerably change.

Second, a major limitation is that the period (6 years) of the analysis is short for attributing the observed trends to anthropogenic changes only, as short-term meteorological variability (not accounted here) might partly explain the observed trends. Furthermore, the fact that the study period includes the COVID-19 lockdowns makes the derivation of trends even more uncertain. I recommend extending the analysis to a longer period, by combining OMI and TROPOMI datasets, both having the same overpass (e.g. Glissenaar et al. https://doi.org/10.5194/essd-2024-616).

Third, the seasonal variation of $NO_2$ data is not adequately discussed. It is unclear whether a threshold of available observations per month was used and why the May-to-September trends are not compatible with the November-to-March trends in some regions, e.g. Europe. Overall, the discussion is poor, and the scientific content is limited. To my opinion, the main problem of the manuscript is the lack of innovation and inadequate analysis. Therefore, I cannot accept this manuscript for publication in ACP.

Thank you for providing this thoughtful feedback. Below, we list the ways in which the above concerns have been addressed. We are confident that the value and quality added to the manuscript in including these substantial changes will be apparent.

1. In this revision, we have performed an uncertainty analysis, being sure to provide statistical significance where relevant. We have made sure to note clearly if a trend is or is not statistically significant. After performing this analysis, our original findings have been strengthened.

2. We do not claim that the observed changes are only anthropogenic in nature; we are simply reporting observed concentrations. Previous work has shown that meteorological variability has lesser influence when aggregating to the country or continental level as opposed to at the local level (Petetin et al., 2020), and our presented evaluation of $NO_2$ VCD urban enhancements against anthropogenic emissions inventories is only presented at the urban continental level. Despite this, even if hypothetical changes in $NO_2$ over a six-year period were to be driven by meteorological as opposed to anthropogenic factors, we still find that such a trend would be worth reporting. We have added the following text to line 510 in the conclusions of the revised manuscript to acknowledge these limitations:

   "Additionally, while many of the trends presented here reflect changes in anthropogenic NOx emissions, it is important to recognize that atmospheric chemistry also influences the observed $NO_2$ variability. Seasonal differences in photochemical lifetimes (i.e., longest in winter), boundary layer mixing (i.e., more vertical mixing in summer), chemical partitioning between NO and $NO_2$ (i.e., the fraction of $NO_2$ is largest in winter) and meteorological variability can all modulate the magnitude and timing of observed trends. These processes likely contribute to some of the regional and seasonal differences highlighted in this study".

3. We appreciate the suggestion to include a longer time series by including an evaluation of OMI trends, however, there are instrument (i.e., coarser resolution) and algorithm differences that make a stitching of these two datasets complex. The inherent advantage of this work is that we are using a single instrument with a consistent algorithm throughout, to calculate the trends. There has been prior work documenting $NO_2$ trends between 2005 – 2019 using OMI and it would be repetitive to repeat that work. We instead cite the OMI $NO_2$ trends papers to give the reader references to refer to.

4. The novelty in this paper is the timeframe and global expansiveness (11,000 cities) of the analysis that we are conducting. To our knowledge, no published paper reports satellite-based $NO_2$ trends through the year 2024. We are documenting urban and country-level $NO_2$ trends that, before this paper, have not been documented in the literature. For example, it has not been documented that Seoul has the largest $NO_2$ drop between 2019 and 2024 and that Tehran has the largest $NO_2$ burden in 2024; this scratches the surface of newly documented $NO_2$ trends that we report herein. This is an important scientific advancement because we are documenting which urban areas appear to be implementing effective control strategies and which are not. This has implications for future control strategies.

5. Thank you for the questions regarding seasonality. We have expanded our discussion in the revised manuscript to more clearly discuss those results. The noted difference between May-September and November-March in Europe is driven by Western Russian cities, which caused the initial spike in Winter 2022. We have added the same figure to the supplement with Russian cities removed (Fig. S17), to show that seasonal trends match more closely when excluding Russian cities.

Specific comments:

- L.112-116: Any data filtering used?

Thank you for allowing us to clarify. We are using a level 3 data product (Goldberg, 2024), which applies a filter to remove all pixels with a qa_value filter < 0.75. We have added the following text to line 122 of the revised manuscript to make this clear:

"Data were quality controlled to remove Level 2 pixels with a qa_value < 0.75 before oversampling, which removes data with quality issues related to clouds, surface reflectivity (e.g. snow and ice) or other retrieval errors".

- L. 57: Read SCIAMACHY

Thank you for pointing this out. We have corrected the error.

- Why not using CEDS (instead of EDGAR)? The latest CEDS version covers 2023.

Thank you for this suggestion. We have now expanded the analysis to include an evaluation of EDGAR as well as CEDS NOx emissions. Results can be found in Section 5 of the revised manuscript.

[Figure]

- A table summarizing the percentage changes would be useful

We appreciate this suggestion, however given the large number of cities (11,000+), we find including a table of all percent changes would not be realistic.

- Figs 2, 4, 5, 6 are similar and could be grouped in 2

Based on this suggestion, we re-grouped some of the figures from the original submission to have one main figure per continent, which more effectively groups results based on geographical location.

- Unclear whether the strong column changes in Quito and Santos are due to changes in anthropogenic activities, or to data issues.

The level 2 TROPOMI data used to produce the level 3 product used in this paper (Goldberg, 2024) are quality controlled to remove all pixels with a QA_flag value < 0.75, which remove pixels with data issues. There is no evidence for the changes in these cities being attributed to "data issues". We do not claim that these changes are a direct result of anthropogenic activities; we are simply highlighting cities that exhibited notable changes in $NO_2$ VCD.

- For which among the cities of Fig.7 are the trends significant?

Thank you for this question. For the labeled cities in each panel, we have noted which represent a statistically significant trend with an asterisk. These results are now spread across Figures 6-11.

References

Goldberg, D.L..: HAQAST Sentinel-5P TROPOMI Nitrogen Dioxide (NO2) GLOBAL Monthly Level 3 0.1 x 0.1 Degree Gridded Data Version 2.4 (HAQ_TROPOMI_NO2_GLOBAL_M_L3) at GES DISC, Goddard Earth Sciences Data and Information Services Center (GES DISC), https://doi.org/10.5067/KKPPL39PEIGE, 2024.

Petetin, H., Bowdalo, D., Soret, A., Guevara, M., Jorba, O., Serradell, K., and Pérez García-Pando, C.: Meteorology-normalized impact of the COVID-19 lockdown upon NO2 pollution in Spain, Atmos Chem Phys, 20, 11119–11141, https://doi.org/10.5194/ACP-20-11119-2020, 2020.

---

## Author Comment (AC2)

Response to Reviewer 1:

The manuscript titled "TROPOMI NO$_2$ for urban and polluted areas globally from 2019 to 2024"
presents a comprehensive analysis on NO$_2$ VCD changes in cities worldwide. It details the
contrasts in NO$_2$ trends across cities, and potential drivers the embedded anthropogenic
emissions, including environmental regulation, local economic growth and regional conflicts.
Although the study illustrates the latest evolution of global air pollution, and offers a valuable
reference for future research, the manuscript, in its current form, contains several critical issues
that warrant major revisions. Therefore, I recommend reconsideration for its publication after the
authors adequately address the concerns outlined below.

We sincerely thank the reviewer for the thoughtful review to improve the quality of this manuscript.

The current manuscript lacks a discussion of the uncertainty of NO$_2$ VCDs and its potential
impacts on the conclusions. This information is crucial for distinguishing trends from interannual
fluctuations, and for separating meaningful emission changes from the noise inherent in satellite
retrievals. However, uncertainty considerations are absent from the main text and figures. In
addition, further validation of the NO$_2$ background values is necessary, along with sensitivity tests
(e.g. evaluating the results using different percentile thresholds in the background selection). The
interannual variability of the background should also be evaluated (e.g. in Fig. 12), as this could
influence the interpretation of relative changes in VCD enhancements. Moreover, the spatial
consistency of the background should be examined, particularly in regions where adjacent cities
are expected to share similar background levels.

We appreciate this feedback, and have made the following additions to the manuscript to address
these concerns:

1.  We have now added statistical significance testing to all trend plots. Significance is
determined through linear regression on monthly de-seasonalized TROPOMI time series
(see revised Sec. 2.3). We successfully show that statistically significant trends can be
separated from insignificant trends from the monthly TROPOMI data from 2019 to 2024,
(e.g. in Figures 2-5), and that this time period is not too short.
2.  To address general uncertainty within TROPOMI retrievals, we added text to note the 15-
20% relative uncertainty related to monthly and annual averages, as highlighted in a
recent previous study (Glissenaar et al., 2025). We also added text to mention the
systemic biases of operational TROPOMI NO$_2$ retrievals, that can lead to underestimated
NO$_2$ VCDs over highly polluted regions (-31.4% bias for values >15x10$^{15}$) and
overestimated (+26.5% bias) NO$_2$ VCDs in less-polluted regions with VCDs <2 x10$^{15}$
(Lambert et al., 2025). To address uncertainty associated with estimates at the city and
country level, we have included uncertainty estimates when reporting relevant values.
3.  We thank the reviewer for the suggestion to re-evaluate the method used to quantify the
background NO$_2$ concentration. We have added a section to the supplementary document
(Section S1 Sensitivity of Urban Background NO2 VCDs) in which we conduct a sensitivity
test to evaluate the impact of using different percentile thresholds on the results related to
the VCD enhancement (previously Figure 12, which has now been split up into multiple
figures). In that supplementary section, we test using different percentile thresholds as the
background concentration and find that changing the used percentile does not
meaningfully impact our results nor change the directionality of the trends. Following this
analysis, we find that the 50$^{th}$ percentile as the threshold is an adequate choice for most cities (See Supplementary information). In that same section we highlight test case of
large adjacent cities and how the background concentrations for those cities vary.
Although more sophisticated methods of background quantification exist (Fioletov et al.,
2025) we find that using a percentile as an assumed background concentration is an
acceptable choice given the large number of cities being evaluated (N >11,000).
4. Based on comments from this and the other reviewers, we have made substantial changes
to the revised manuscript, by rearranging many of the figures and sections for clarity. We
are confident these changes greatly improve the quality of this work.

The manuscript includes several qualitative descriptions that are not supported by sufficient
validation or statistical testing. For instance, it states that there is an accelerated decreasing trend
in $NO_2$ VCDs in both China and European countries. However, given that the dataset used in this
study begins in 2019, the time range may be too short to detect or validate such trend
acceleration. Similarly, the manuscript mentions an accelerated $NO_2$ increase over Moscow in
early 2022. Yet, Fig. S9 appears to show only a brief, anomalous spike in $NO_2$ VCDs, followed by
a return to typical levels. These interpretations, as currently presented, are questionable and
require rigorous statistical validations.

We have greatly expanded the statistical testing within this iteration of the manuscript and have
removed any claims of "accelerating" trends. Significance is determined through linear regression
on de-seasonalized TROPOMI time series (see revised Sec. 2.3). We successfully show that
statistically significant trends can be separated from insignificant trends from the monthly
TROPOMI data from 2019 to 2024 (e.g. in Figures 2-5), and that this time period is not too short.

The manuscript appears to insufficiently account for the effects of seasonality on $NO_2$ VCDs.
Given the strong seasonal variation in NOx lifetime, particularly the longer lifetime during winter,
$NO_2$ VCDs in colder months can disproportionately influence interannual trends if seasonality is
not properly addressed. However, the manuscript lacks adequate discussion or correction for
these seasonal effects. Moreover, there appears to be a mischaracterization of seasons between
the Northern and Southern Hemispheres. For instance, the manuscript uses data from the same
calendar months to represent winter conditions in both Asia and Oceania. This approach is
problematic, as most cities in Oceania are located in the Southern Hemisphere, where the
seasonal cycle is inverted. As a result, the analysis may misrepresent seasonal trends in these
regions, and further clarification or adjustment is necessary.

We thank the reviewer for addressing the question of seasonality in our work. To address these
points, we have:

1. Replaced any existing monthly time series within this study with de-seasonalized trends.
This was done for Figure 12 (previously Figure 3) and Figure 3 (previously Figure 8).
2. We have now included a more robust analysis of statistical significance throughout the
manuscript. We use the de-seasonalized data to quantify significance of any trends.
3. We have addressed the discrepancy between Northern and Southern hemispheric cities
by removing and reference to "warmer months" or "colder months" or season names,
particularly when discussing South America, Africa or the lumped Asia and Oceania
section. We instead refer to the months of observations, e.g. May-September.

**Specific Comments**

Page 2, Line 56-67: I would suggest to include a brief overview about $NO_2$ VCD changes in India, Oceania and Africa here, since these regions also play important roles in this study.

We appreciate this suggestion. We have added the following text at line 67 of the revised manuscript:

"In contrast, urban regions of India have shown $NO_2$ increases over the past few decades, linked to urbanization and energy demand growth (Hilboll et al., 2013; Ghude et al., 2020). Over Africa and South America, $NO_2$ VCD trends through the mid-2010s have been less pronounced, reflecting limited industrialization and more dominant contributions from biomass burning and natural sources (Geddes et al., 2016; Castellanos et al., 2014)."

Page 2, Line 59: "x" --> "×". Check throughout the manuscript.

Thank you for pointing this out. We have made the change to all mentions within the manuscript as well as any figures using the notation.

Page 4, Line 97: It should be explained why GHS-SMOD boundaries are used rather than administrative city boundaries, and clarify whether this choice affects the results.

Thank you for this question. To our knowledge, a dataset of true administrative / legislatively determined boundaries for all urban regions globally does not exist. We have added the following text at line 102 of the revised manuscript to emphasize the value of using GHS-SMOD:

"GHS-SMOD has the benefit of providing a globally consistent, satellite-derived definition of built-up areas, whereas administrative boundaries vary widely in definition and availability. Using physical built-up area boundaries from GHS-SMOD instead of administrative ones may shift the absolute spatial extent of some cities, but it does not materially alter the concentrations calculated in this study."

Page 4, Line 111: According the latest ATBD (2.8.0, 2024-11-18– released) for TROPOMI $NO_2$, the nadir ground pixel dimensions were $7.0 \times 3.5$ km$^2$ before 6 August 2019. The data description here is inaccurate.

Thank you for pointing this out. We have modified the text at line 121 of the revised manuscript to reflect the difference in spatial resolution before 06 August, 2019:

"These Level 2 data have a nadir spatial resolution of $3.5 \times 7.0$ km$^2$ before and $3.5 \times 5.5$ km$^2$ after 06 August 2019."

Page 6, line 142: Sensitivity tests should be conducted to assess the impact of using different percentile values in background selection. In addition, validation is needed. For example, by examining whether background values are consistent across adjacent cities.

We appreciate this suggestion. We have added a section to the supplement titled "S1 Sensitivity of Urban Background $NO_2$ VCDs", where we (1) describe the method used to quantify background concentrations, (2) provide results of a sensitivity analysis evaluating the impact of different percentiles on the general conclusions we've drawn within this manuscript, (3) an evaluation of trends in urban background VCDs over time and (4) the impact and evaluation of background for adjacent cities / clusters of cities. In short, following this thorough analysis, we have updated the choice of background concentration to the 50th percentile. We test using different percentile thresholds as the background concentration and generally find that changing the used percentile does not meaningfully impact our results nor change the directionality of the trends. Please refer to the revised methods section (Sec. 2.4) and the supplement document for more information.

Page 6, Line 157: The claimed acceleration in the decreasing trend requires statistical validation; otherwise, such descriptions might be just removed. (Also, for the descriptions on Page 10, Line 239, Page 10, Line 221, and Page 14, Line 310)

Based on this recommendation, we have changed the phrasing of this specific text to no longer claim an accelerated trend, but rather a continued decreasing trend. We note that the statistical analysis in this revised submission has been greatly expanded, and there are now many references to statistical significance of trends throughout this work.

Page 7, Line 164: Please clarify the definition of the mining regions (including A, C in Fig. 2; B in Fig. 4; D, E in Fig. S4; and G, F, H, I in Fig. 6).

We have added the description of the mining regions within the captions of each figure (Figures 6-11 in the revised manuscript)

Page 7, Line 166: The texts in Fig. S3 are not clear.

We have increased the text size in the legends in the top left of each panel. This is now Figure S20.

Page 8, Figure 3: The information of $NO_2$ VCD uncertainty and significance tests on the regression is missing. In addition, please ensure consistency of significant figures or decimal precision for all numerical data throughout the manuscript.

Thank you for this suggestion. We have now changed this figure to show de-seasonalized trends as a monthly % anomaly, from which trends can more clearly be quantified and visualized. For each panel, we have added the statistical significance of the trend by reporting a percent change per year, as well as a p-value. We have also updated the manuscript to ensure the use of consistent significant figures. See Sec. 2.3.

Page 9, Line 197: Please provide the specific number and proportion ("Nearly all").

Thank you for this suggestion. We have since removed this text for the sake of shortening the manuscript. However, we calculated that 66 of the 71 urban clusters in Eastern Russia (or 93%) exhibited larger 2024 $NO_2$ VCDs than in 2019.

Page 9, Figure 4: I would suggest standardizing the formatting of units throughout the manuscript for consistency.

We have double checked the manuscript to ensure that the units (e.g. $10^{15}$ Molecules cm$^{-2}$) are used consistently throughout the manuscript. If this is in reference to the range of plotted data on the colorbars (e.g. -1 to1 in Fig. 4 vs. -2 to 2 in Fig. 6), we find that different regions require
different values on the color bar for the sake of effective data visualization, as not all regions
experience a similar range of concentrations.

Page 10, Line 232: What is the term "largest" referring to or being compared against in this
description? (other cities or other land type? Also, for the descriptions on Page 11, Line 248, Page
11, Line 253-254, Page 12, Line 263-264 and Page 12, Line 270)

These are directly in reference to the largest observed annual mean concentrations or differences
in the spatial figures (Now Figures 6-11 in Sections 5.1 – 5.5; largest concentration refers to data
plotted in panel a, while largest increase/decrease refers to data plotted in panel b). In the case
of Europe as is referenced here, we are claiming that the largest values occur in urban settings,
as opposed to non-urban settings (See Fig. 8a). In reference to Moscow, we are stating that the
largest concentrations in Europe were observed over Moscow. Hypothetically, a power plant, or
coal mine or some other source outside of the urban environment could in theory be the largest
signature for a region. We are simply noting that the largest signatures observed here are in urban
environments. We have made sure that these figures are properly referred to in the revised
manuscript.

Page 10, Line 236: Please provide the specific number.

We have added the modified the text on line 354 of the revised manuscript, which now reads:

"Of the 1257 urban clusters in Europe, 1007 (80%) exhibited larger VCDs in 2024 than in 2019.
Of the 53 European urban clusters with a population greater than 1,000,000, 2024 VCDs were
lower than 2019 VCDs in 48 (91%), with the exception of Moscow and other cities of western
Russia, which experienced increases (Fig. 9b)"

Page 12, Section 4: I would suggest to integrate Section 3 and Section 4.

Thank you for this suggestion. We have largely restructured the manuscript by combining multiple
sections, although the content generally remains the same. Section 3 now highlights trends in
major urban areas, Section 4 quantifies country-level, population-weighted trends, Section 5
highlights continental signatures and seasonality, and Section 6 highlights trends in major oil, gas
and metal mining regions.

Page 13, Figure 7: The figure legend could be further improved to enhance readability.

We have increased the size of the legend for readability.

Page 14, Line 311: Typo.

Thank you, we have fixed the typo.

Page 14, Line 311: The abnormally high $NO_2$ VCD values require further examination to exclude
artifacts, including applying data filters based on Level-2 QA flags. It should also be verified
whether any spurious outliers affect the averaging process.

The oversampled level 3 TROPOMI $NO_2$ product we use in this study are quality controlled, with
the recommended QA flag > 0.75 applied to the L2 data before oversampling. It removes cloud-
covered scenes (cloud radiance fraction> 0.5), partially snow/ice covered scenes, errors, and
problematic retrievals. The number of valid observations for the Moscow urban cluster during
March 2022, when a monthly average value of $59 \times 10^{15}$ molec. $cm^{-2}$ was calculated, had more
observations (233) than the median March (159) in the six-year record. Despite the abnormally
high value, this appears to be a valid mean value for that month.

Page 15, Figure 8: The figure labels/text are not clear.

We have increased the size of the text for readability.

Page 17, Line 349-350: Such causal relationships require careful validation. I recommend revising
the statement here.

Thank you for pointing this out. Numerous studies have identified the impact of the shutdowns in
China on VCDs within the country, which we cite on line 274 of the revised manuscript (Zheng et
al., 2021; Cooper et al., 2022; Levelt et al., 2022; Ma et al., 2023; Zhao et al., 2024), to make
clear this is a reference to previous studies and not determined from our work alone.

Page 18, Line 376: It is not immediately clear why population-weighted VCDs are preferred here
over direct $NO_2$ VCDs for me. Would directly showing $NO_2$ VCDs make major differences?

We choose to use population-weighted VCDs when aggregating by region (continent or country),
as it more accurately reflects the column concentration that people in a region's urban areas are
exposed to. An equitable spatial average would give unnecessary weight to concentrations in less
populous cities where fewer people live and the relative satellite uncertainties are larger.

Page 21, Line 433: Since the comparison here is based on the relative changes of VCDs and
emissions with respect to 2019, it is hard to conclude that emissions are underestimated. At most,
it may suggest a possible underestimation in the emission trend. (Also for Page 24, Line 481)

We apologize for the unclear language. We never meant to say that the emissions are
underestimated in totality but instead underestimated trends. We have modified the text to reflect
the year-to-year variability (i.e. trend) in the emissions inventory is likely underestimated based
on discrepancies between its trends and TROPOMI's trends now on line 340:

"Evaluating trends in NOx emissions inventories in African cities, we find a mean difference of -
8.0% (EDARv8.1) and -6.7% (CEDS) between inventory NOx emission trends and $VCD_{ENH}$
trends, indicating a potential underestimate in both emissions inventories in African cities for this
period."

Page 21, Line 435: Impacts of uncertainty in VCD background need to be quantified.

We thank the reviewer for this suggestion. We have added a section to the revised supplement
document that provides an analysis of the urban background concentrations. Please refer to the
revised supplement document Section S1.

Page 21, Line 435: The mean difference is likely underestimated due to the inclusion of 2019.

Thank you for pointing this out. As 2019 has a value of 0, we have ensured that data from that
year are not included in the average, and we have updated the relevant values.

Page 22, Line 451: There appears to be a mischaracterization of seasons between the Northern
and Southern Hemispheres, since Asia and Oceania are shown together in Fig. 13.

Thank you for noting this. We have modified the text throughout the manuscript to refer to the
months explicitly (e.g. May-September), as opposed to "warmer months" or "summer", particularly
when referring to South America, Africa and Oceania.

Page 23, Figure 13: Is the sharp increase during the winter of 2022 primarily driven by
anomalously high values over Russia? If so, the authors should consider presenting additional
results with Russia excluded. Intuitively, I find that this sharp increase appears inconsistent with
Fig. 9c, where most cities do not show a similar increase in 2022.

Thank you for bringing up this point. The sharp increase observed in winter 2022 in Europe is
indeed due to increases in Russian cities. We have created a version of this figure with Russian
cities removed and added to the supplement as Figure S18, and added the following text to line
373 of the revised/tracked changes manuscript:

"We note that the seasonal trends in Europe show notable winter and summer decrease if
evaluating trends with Russian cities removed (Fig. S18)."

The noted difference in the magnitude for Europe in 2022 between previous Figure 13 and Figure
9 is due to the fewer data points during winter. When averaged over the colder months, the 2022
increase is stark, as seen in Figure 13. When averaging to an annual value, the higher number of
observations in the warmer months, when an increase was not observed, dilutes the impact of
the winter values.

Page 24: Line 481: Discussion about the impacts of NOx chemistry and its seasonality should be
included.

Thank you for this suggestion. We have added the following text to line 510 of the revised
manuscript:

"Additionally, while many of the trends presented here reflect changes in anthropogenic NOx
emissions, it is important to recognize that atmospheric chemistry also influences the observed
$NO_2$ variability. Seasonal differences in photochemical lifetimes (i.e., longest in winter), boundary
layer mixing (i.e., more vertical mixing in summer), chemical partitioning between NO and $NO_2$
(i.e., the fraction of $NO_2$ is largest in winter) and meteorological variability can all modulate the
magnitude and timing of observed trends. These processes likely contribute to some of the
regional and seasonal differences highlighted in this study."

Page 24, Line 491-492 ("tall-stack sources"): Could the authors provide supporting references for
this statement?

We have added the citation to line 506 (Brett et al., 2025).

References

Brett, N., Arnold, S. R., Law, K. S., Raut, J.-C., Onishi, T., Barret, B., Dieudonné, E., Cesler-Maloney, M., Simpson, W., Bekki, S., Savarino, J., Albertin, S., Gilliam, R., Fahey, K., Pouliot, G., Huff, D., and D'Anna, B.: Estimating Power Plant Contributions to Surface Pollution in a Wintertime Arctic Environment, ACS ES&T Air, 2, 943–956, https://doi.org/10.1021/ACSESTAIR.5C00030, 2025.

Cooper, M. J., Martin, R. V., Hammer, M. S., Levelt, P. F., Veefkind, P., Lamsal, L. N., Krotkov, N. A., Brook, J. R., and McLinden, C. A.: Global fine-scale changes in ambient NO2 during COVID-19 lockdowns, Nature 2022 601:7893, 601, 380–387, https://doi.org/10.1038/s41586-021-04229-0, 2022.

Fioletov, V., Mclinden, C. A., Griffin, D., Zhao, X., and Eskes, H.: Global seasonal urban, industrial, and background NO2 estimated from TROPOMI satellite observations, Atmos. Chem. Phys, 25, 575–596, https://doi.org/10.5194/acp-25-575-2025, 2025.

Glissenaar, I., Boersma, K. F., Anglou, I., Rijsdijk, P., Verhoelst, T., Compernolle, S., Pinardi, G., Lambert, J.-C., Van Roozendael, M., and Eskes, H.: TROPOMI Level 3 tropospheric NO 2 dataset with advanced uncertainty analysis from the ESA CCI+ ECV precursor project , Earth Syst Sci Data, 17, 4627–4650, https://doi.org/10.5194/ESSD-17-4627-2025, 2025.

Lambert, J.-C. et al.: Quarterly Validation Report of the Copernicus Sentinel-5 Precursor Operational Data Products #28: April 2018 – August 2025, https://mpc-vdaf.tropomi.eu/ProjectDir/reports/pdf/S5P-MPC-IASB-ROCVR-28.00.00_FINAL_signed-jcl-AD.pdf, 2025.

Levelt, P. F., Stein Zweers, D. C., Aben, I., Bauwens, M., Borsdorff, T., De Smedt, I., Eskes, H. J., Lerot, C., Loyola, D. G., Romahn, F., Stavrakou, T., Theys, N., Van Roozendael, M., Veefkind, J. P., and Verhoelst, T.: Air quality impacts of COVID-19 lockdown measures detected from space using high spatial resolution observations of multiple trace gases from Sentinel-5P/TROPOMI, Atmos Chem Phys, 22, 10319–10351, https://doi.org/10.5194/ACP-22-10319-2022, 2022.

Ma, Q., Wang, J., Xiong, M., and Zhu, L.: Air Quality Index (AQI) Did Not Improve during the COVID-19 Lockdown in Shanghai, China, in 2022, Based on Ground and TROPOMI Observations, Remote Sens., 15, 1295, https://doi.org/10.3390/RS15051295/S1, 2023.

Zhao, X., Li, X. X., Xin, R., Zhang, Y., and Liu, C. H.: Impact of Lockdowns on Air Pollution: Case Studies of Two Periods in 2022 in Guangzhou, China, Atmos. 2024, Vol. 15, Page 1144, 15, 1144, https://doi.org/10.3390/ATMOS15091144, 2024.

Zheng, B., Zhang, Q., Geng, G., Chen, C., Shi, Q., Cui, M., Lei, Y., and He, K.: Changes in China's anthropogenic emissions and air quality during the COVID-19 pandemic in 2020, Earth Syst. Sci. Data, 13, 2895–2907, https://doi.org/10.5194/ESSD-13-2895-2021, 2021.

---

## Author Comment (AC3)

**General comments:**

This paper presents a comprehensive assessment of urban $NO_2$ changes worldwide from 2019 to 2024 using TROPOMI $NO_2$ VCD observations. Differences in $NO_2$ VCD changes over populous cities and broader areas are disclosed, probably driven by anthropogenically induced factors such as urbanization, industrial activities, government interventions, and societal disruptions. The paper also attempts to quantify the influence of background $NO_2$ and $NO_2$ seasonal variability on the trend analysis. The research topic fits in the scope of ACP, and the manuscript is already in good shape. I recommend its publication after the authors address the following comments.

We thank the reviewer for this thoughtful review.

Based on comments from this and the other reviewers, we have made substantial changes to the revised manuscript, by rearranging many of the figures and sections for clarity. We are confident that these changes greatly improve the quality of this work.

**Specific comments:**

Line 1-2: The current title is a little general and can not convey the key point of this research. I would suggest to improve the title by using the key conclusion of this study, which can better draw readers' attention.

We appreciate this suggestion. We have changed the title of the manuscript to "Global $NO_2$ Trends from TROPOMI (2019–2024): Urban Changes and Emerging Hotspots".

Line 39-40: The remote sensing method not only relies on spectrometers aboard satellites to infer vertical columns, but also can infer vertical profiles using ground-based instruments. The statement should incorporate the profile retrieval to ensure a more comprehensive description.

Thank you for this suggestion. We have modified the text on line 44 of the revised manuscript to reflect the different types of remote sensing methods that exist for $NO_2$:

"$NO_2$ can also be remotely-sensed from ground-based instruments capable of inferring vertical profiles of $NO_2$, such as using multi-axis differential optical absorption spectroscopy, MAX-DOAS (Vlemmix et al., 2010)"

Line 68-75: This paragraph only describes the different methods used to characterize the urban extent. The authors should add a brief discussion about the pros and cons of these methods, and highlight the advantage(s) of the GHS-SMOD, which is used in this study.

Thank you for this suggestion. We have added the following text to the methods section (Line 102) to highlight the value of GHS-SMOD:

"GHS-SMOD has the benefit of providing a globally consistent, satellite-derived definition of built-up areas, whereas administrative boundaries vary widely in definition and availability. Using physical built-up area boundaries from GHS-SMOD instead of administrative ones may shift the absolute spatial extent of some cities, but it does not materially alter the concentrations calculated in this study."

Section 2.2: Please briefly describe the uncertainty of the $NO_2$ VCD product used here.

Thank you for this suggestion. To address general uncertainty within TROPOMI retrievals, we added text to note the 15-20% relative uncertainty related to monthly and annual averages, as highlighted in a recent previous study (Glissenaar et al., 2025). We also added text to mention the systemic biases of TROPOMI $NO_2$ retrievals, that can lead to underestimated $NO_2$ VCDs over highly polluted regions, and overestimated $NO_2$ VCDs in less-polluted regions (Lambert et al., 2025). To address uncertainty associated with estimates at the city and country level, we have added error bars to relevant figures and include uncertainty estimates when reporting relevant values.

We have added text to line 129 of the revised manuscript, highlighting the most common areas of uncertainty related to $NO_2$ retrievals.

"TROPOMI $NO_2$ retrievals are subject to measurement and retrieval uncertainties that propagate into the oversampled Level 3 products. Typical uncertainties in monthly or annually averaged tropospheric $NO_2$ vertical column densities are on the order of 15–20 %. Systematic biases have also been reported, with overestimation in less polluted regions (+26.5% bias) and underestimation in areas with high $NO_2$ concentrations (-31.4% bias), reflecting limitations in the retrieval process (Glissenaar et al., 2025; Lambert et al., 2025)."

Additionally, we have now added statistical significance testing to all trend plots. Significance is determined through linear regression on monthly de-seasonalized TROPOMI time series (see revised Sec. 2.3). We successfully show that statistically significant trends can be separated from insignificant trends from the monthly TROPOMI data from 2019 to 2024, (e.g. in Figures 2-5).

Section 2.2.1: The structure here is a little inappropriate because there is only one sub-section. I would suggest to merge Section 2.2.1 and Section 2.2 to one section.

We have removed subsection 2.2.1 and turned it into a separate section 2.3 and changed existing sections 2.3 and 2.4 to 2.4 and 2.5, respectively.

Line 141-142: Please justify the definition of the background $NO_2$ concentration here, and provide the sensitivity of the results in Section 6 to the choice of the percentile.

Thank you for this suggestion. We have added a section to the supplementary document (Section S1 Sensitivity of Urban Background NO2 VCDs) in which we conduct a sensitivity test to evaluate the impact of using different percentile thresholds on the results related to the VCD enhancement (previously Figure 12, which has now been split up into multiple figures). In that supplementary section, we test using different percentile thresholds as the background concentration and find that changing the used percentile does not meaningfully impact our results nor change the directionality of the trends. Following this analysis, we find that the $50^{th}$ percentile as the threshold is an adequate choice for most cities (See Supplementary information). In that same section we highlight test case of large adjacent cities and how the background concentrations for those cities vary. Although more sophisticated methods of background quantification exist (Fioletov et al., 2025), we find that using a percentile as an assumed background concentration is an acceptable choice given the large number of cities being evaluated (N >11,000).

Section 2.4: Please briefly describe the uncertainty of the EDGARv8.1 NO$_x$ emissions.

At the suggestion of a separate reviewer, we have included an evaluation against CEDS as well as EDGAR. We have added the following text to line 176 of the revised manuscript (now in Section 2.5) describing general uncertainties in these bottom-up inventories:

"Uncertainties are inherent in such emissions inventories, with a roughly 10-50% uncertainty when aggregating emissions to the country level, and even larger uncertainty for individual grid points (Crippa et al., 2018).".

Line 157-158: Is the statement "the decrease accelerated after the onset of the COVID-19 pandemic" one of the findings of this study, or a knowledge cited from other papers? If the former is true, please provide a quantitative discussion to support this point; if the latter is true, please provide supporting references.

Thank you for allowing us to clarify this point. We have modified the text to note "a continued decreasing trend", as opposed to an accelerated trend.

Figure 3, Figure 8 and Figure S6: please provide the confidence level of each regression to clarify the statistical significance of the characterized trends.

Thank you for this suggestion. For each of these monthly level examples, we have now added statistical significance testing to all trend plots. Significance is determined through linear regression on monthly de-seasonalized TROPOMI time series (see revised Sec. 2.3). We successfully show that statistically significant trends can be separated from insignificant trends from the monthly TROPOMI data from 2019 to 2024, (e.g. in Figures 2-5).

Line 305-306: Please provide a quantitative discussion to support the statement "The observed annual decreases in these East Asian cities were primarily driven by decreases during the winter months".

We have removed this sentence from the revised manuscript.

Line 310: It is difficult to see that the increasing trend in Moscow accelerated in early 2022 from Figure S9, except that NO$_2$ VCDs in winter time jumped to a higher level. Please provide a quantitative discussion to demonstrate the acceleration.

That is a good point. We have changed the text to no longer claim that the trend is accelerating, now on line 215 of the revised manuscript.

"This increase was accompanied by anomalously high monthly mean concentrations in early 2022 (Fig. S11), following the onset of the Russia-Ukraine war in Ukraine, when monthly mean NO$_2$ VCDs for March reached 59 × 10$^{15}$ molecules cm$^{-2}$ (see Sec. 3.3)"

Section 6: please provide a summary of this section, i.e., to what extent the influence of background $NO_2$ and seasonal variability can be on the analysis of urban $NO_2$ trends presented above?

Thank you for this suggestion. We have added a section to the supplementary document (Section S1 Sensitivity of Urban Background NO2 VCDs) in which we conduct a sensitivity test to evaluate the impact of using different percentile thresholds on the results related to the VCD enhancement (previously Figure 12, which has now been split up into multiple figures). In that supplementary section, we test using different percentile thresholds as the background concentration and find that changing the used percentile does not meaningfully impact our results nor change the directionality of the trends. Following this analysis, we find that the 50th percentile as the threshold is an adequate choice for most cities (See Supplementary information). In that same section we highlight test case of large adjacent cities and how the background concentrations for those cities vary. Although more sophisticated methods of background quantification exist (Fioletov et al., 2025), we find that using a percentile as an assumed background concentration is an acceptable choice given the large number of cities being evaluated (N >11,000).

Line 446-456 and line 482: It should be careful to define May – September and November – March as either "warm" or "cold" months, given the different hemispheres in which the continents are located. The interpretation of the results for Asia and Oceania is problematic, because May – September is summer time for Asia but is winter time for Oceania, while November – March is winter time for Asia but is summer time for Oceania. Please revise the discussions here and in Section 7.

Thank you for this suggestion. We have modified the text related to time of year by only referring to the months used (e.g. May-September) as opposed to referring to a time of year as warmer or colder.

**Technical comments:**

Line 50: Please check through the manuscript and replace the "x" with a times symbol at corresponding places.

We have replaced all "x" with an "×" symbol both within the text and in each relevant figure.

Line 57: "SCHIAMACY" should be "SCIAMACHY".

We have fixed the error (now line 49).

Line 63: The statement "$NO_2$ concentrations increased through roughly 2005" is a little confusing. I would suggest to rephrase this sentence to make it clearer.

We have modified the text on line 65 of the revised manuscript, which now reads:

"In the United States, $NO_2$ concentrations generally exhibited a decreasing trend from 2005 through the mid-2010s…".

Line 94: It is better to be $1 \times 1$ $km^2$.

We have made the change (now line 100).

Line 127: Please add a period after "approach".

Thank you for catching this. We have added the period.

Line 142: It is clearer to extend "UC" to "urban cluster".

Thank you, we have made that change.

Line 148: Do you mean "Sec. 2.2.1" here?

That is correct, thank you. We have since modified this to read "Sec. 2.3".

Line 183: "select" to "selected".

We have made the change to the Figure 3 caption.

Line 306: There is no Figure 7d.

Thank you for catching this. This was supposed to read Figure 8d, not 7d. We have changed this in the text.

References

Crippa, M., Guizzardi, D., Muntean, M., Schaaf, E., Dentener, F., Van Aardenne, J. A., Monni, S., Doering, U., Olivier, J. G. J., Pagliari, V., and Janssens-Maenhout, G.: Gridded emissions of air pollutants for the period 1970-2012 within EDGAR v4.3.2, Earth Syst Sci Data, 10, 1987–2013, https://doi.org/10.5194/ESSD-10-1987-2018, 2018.

Fioletov, V., Mclinden, C. A., Griffin, D., Zhao, X., and Eskes, H.: Global seasonal urban, industrial, and background NO2 estimated from TROPOMI satellite observations, Atmos. Chem. Phys, 25, 575–596, https://doi.org/10.5194/acp-25-575-2025, 2025.

Glissenaar, I., Boersma, K. F., Anglou, I., Rijsdijk, P., Verhoelst, T., Compernolle, S., Pinardi, G., Lambert, J.-C., Van Roozendael, M., and Eskes, H.: TROPOMI Level 3 tropospheric NO 2 dataset with advanced uncertainty analysis from the ESA CCI+ ECV precursor project , Earth Syst Sci Data, 17, 4627–4650, https://doi.org/10.5194/ESSD-17-4627-2025, 2025.

Vlemmix, T., Piters, A. J. M., Stammes, P., Wang, P., and Levelt, P. F.: Retrieval of tropospheric NO2 using the MAX-DOAS method combined with relative intensity measurements for aerosol correction, Atmos Meas Tech, 3, 1287–1305, https://doi.org/10.5194/AMT-3-1287-2010, 2010.